# Permutation Equivariant Neural Controlled Differential Equations for Dynamic Graph Representation Learning

**Torben Berndt**[1][*]    **Benjamin Walker**[2]    **Tiexin Qin**[3]
**Jan Stühmer**[1,4]    **Andrey Kormilitzin**[5]

[1]Heidelberg Institute for Theoretical Studies, Heidelberg, Germany
[2]Mathematical Institute, University of Oxford, UK
[3]City University of Hong Kong, Hong Kong
[4]IAR, Karlsruhe Institute of Technology, Karlsruhe, Germany
[5]Department of Psychiatry, Warneford Hospital, Oxford, UK

## Abstract

Dynamic graphs exhibit complex temporal dynamics due to the interplay between evolving node features and changing network structures. Recently, Graph Neural Controlled Differential Equations (Graph Neural CDEs) successfully adapted Neural CDEs from paths on Euclidean domains to paths on graph domains. Building on this foundation, we introduce *Permutation Equivariant Neural Graph CDEs*, which project Graph Neural CDEs onto permutation equivariant function spaces. This significantly reduces the model's parameter count without compromising representational power, resulting in more efficient training and improved generalisation. We empirically demonstrate the advantages of our approach through experiments on simulated dynamical systems and real-world tasks, showing improved performance in both interpolation and extrapolation scenarios.

## 1 Introduction

Graph Neural Networks (GNNs) [57, 56, 38, 63] have emerged as a leading framework for modelling graph-structured data, demonstrating significant success in applications such as protein folding [31], social recommender systems [22] or traffic forecasting [30]. However, real-world graphs are often dynamic. Protein-protein interactions vary over time due to cellular processes, social networks evolve as relationships shift, and roads close due to building works and car crashes. Effectively capturing these temporal dynamics is crucial for accurate modelling and robust predictions.

For the last century, differential equations have been the cornerstone of modelling continuous change. However, in recent years, deep learning has revolutionised data analysis with its ability to learn complex patterns from vast amounts of data. While these approaches initially developed in parallel, the concept of Neural Differential Equations (NDEs) [46, 52, 7, 34] has bridged the gap, demonstrating their interconnectedness.

Building upon the approach introduced in [48], we propose *Permutation Equivariant Neural Graph Controlled Differential Equations* (PENG-CDEs), a novel framework which addresses key limitations of prior models.[2] Our primary contributions are as follows:

---

[*]Work done in part while at the University of Oxford.

[2]The source code is available at: `https://github.com/hits-mli/perm-equiv-graph-neural-cdes`

39th Conference on Neural Information Processing Systems (NeurIPS 2025).

- Motivated by temporal and spatial symmetries, we derive PENG-CDEs from first principles. Our framework strikes a balance between expressiveness and parameter efficiency (Section 3).

- We formalise this trade-off and prove that, under simplified assumptions, our proposed model is the optimal approximation to Graph Neural CDEs within the space of permutation-equivariant functions (Section 3.2, Theorem 3.1).

- We further prove that the resulting models are equivariant under both time reparametrisations and permutations of the node set (Section 3.2, Proposition 3.2).

- We empirically validate the effectiveness of PENG-CDEs on dynamic node- and graph-level tasks, consistently outperforming both differential equation-based and other spatio-temporal baselines. In particular, our model sets a new state-of-the-art on the TGB-genre node affinity prediction task (Section 4).

## 1.1 Related work

**Continuous-depth GNNs.** Recent work has linked GNNs to continuous-time dynamics on graphs by interpreting hidden layers as a discretised time axis. [6, 54] study convolutional GNNs as discretisations of heat diffusion processes on graphs, while [18] proposes architectures that incorporate both parabolic and hyperbolic PDE terms. Reaction-diffusion dynamics are explored in [12, 17], and advection terms are added in [19]. Higher-order temporal derivatives are considered in [20, 21]. [47] proposes a graph-based formulation of Neural ODEs [7], and Neural SDEs are extended to graph domains in [2]. While these models typically assume static graphs, some work addresses dynamic settings. [11, 10] and [48] extend Neural CDEs to graph domains, by modelling node-level time series and graph structure hierarchically, and by treating the evolving topology as a control, respectively.

**Spatio-temporal GNNs.** A well-established line of work in dynamic graph representation learning models spatial and temporal dynamics using separate modules for each. DCRNN [40] combines a diffusion-based GCN [38] with a recurrent GRU [9] in an encoder-decoder architecture. STGCN [67] applies GCN [16] for spatial modelling and 1D CNNs for temporal processing. Attention-based extensions such as ASTGCN [26] and ASTGNN [27] introduce distinct spatial and temporal attention mechanisms. WaveNet-GCN [42] fuses graph and temporal convolutions, while STIDGCN [41] uses interactive learning to handle heterogeneous time scales.

**Temporal Graph Networks.** Another approach focuses on event-based message passing over time-stamped interactions. TGN [53] introduced this framework, later extended by TGNv2 [59] which considers an identification between source and target nodes. Node-centric methods such as DyRep [60] and TCL [65] compute embeddings from temporal neighbourhoods and aggregate across edges. In contrast, edge-centric methods like CAWN [66] and GraphMixer [14] embed edge interactions directly for prediction.

## 2 Background

### 2.1 (Temporal) Graph representation learning

We denote the set $\{1, \ldots, n\}$ by $[n]$ and the set of functions from $X$ to $Y$ by $\chi(X, Y)$.

**Graphs.** A *graph* $G = (\mathcal{V}, \mathcal{E})$ consists of a collection of *nodes* $\mathcal{V} = [n]$ and *edges* $\mathcal{E} \subseteq \mathcal{V} \times \mathcal{V}$. The graph topology is described by an *adjacency matrix* $\mathbf{A} \in \mathbb{R}^{n \times n}$, where $\mathbf{A}^{ij} = 1$ if $(i, j) \in \mathcal{E}$ and 0 otherwise. The *degree* $d_i = \sum_{j \in \mathcal{V}} \mathbf{A}^{ij}$ of a node is the number of edges incident on it. Let $\mathbf{D}$ be the diagonal *degree matrix* with $\mathbf{D}^{i,i} = d_i$. The *graph Laplacian* is defined as $\mathbf{L} = \mathbf{D} - \mathbf{A}$ and the *normalised graph Laplacian* is defined as $\mathcal{L} = \mathbf{I}_n - \mathbf{D}^{-\frac{1}{2}} \mathbf{A} \mathbf{D}^{-\frac{1}{2}}$ where $\mathbf{I}_n$ is the $n \times n$ identity matrix. We assume the nodes are equipped with $d_x$-dimensional *node features* $\mathbf{x}_i \in \mathbb{R}^{d_x}$, stacked to form a node attribute matrix $\mathbf{X} \in \mathbb{R}^{n \times d_x}$. The goal of *Graph Representation Learning* is to learn a latent node representation $\mathbf{Z} \in \mathbb{R}^{n \times d_z}$ which embeds the nodes into some Euclidean space.

**Temporal Graph Representation Learning.** In this setting, we observe a sequence of graph snapshots $\mathcal{G} = ((t_0, G_{t_0}), \ldots, (t_N, G_{t_N}))$ generated by some unknown continuous-time underlying process. Each snapshot $(t_k, G_{t_k})$ captures the graph state at time $t_k$, with $G_{t_k} = (\mathcal{V}, \mathcal{E}_{t_k})$ and

corresponding adjacency matrices $\mathbf{A}_{t_k} \in \mathbb{R}^{n \times n}$. The objective is to learn a dynamic non-linear latent representation $\mathbf{Z}_t \in \mathbb{R}^{n \times d_z}$ for the nodes for all times $t \in (t_0, t_N]$.

**Equivariance**. Let $G$ be a group and $X, Y$ be two sets with a (left) $G$-action. A function $f : X \to Y$ is $G$-*equivariant* if $f(g \cdot x) = g \cdot f(x)$ for all $g \in G$ and $x \in X$. If the action is trivial (i.e., $g \cdot y = y$ for all $g \in G$ and $y \in Y$), then $f$ is called *invariant*, meaning $f(g \cdot x) = f(x)$ for all $g \in G$ and $x \in X$.

In the context of static graph representation learning, permutation equivariance ensures that the model's output does not depend on the arbitrary ordering of nodes. Time-warp equivariance is important in time series modelling when the task depends on the sequential structure of the signal rather than the absolute timing of events. For example, in classifying the nodes of a social media network as to whether they are friends with a specific node or friends-of-a-friend with that node, the important characteristic is the order in which the edges appeared (you cannot connect with a friend-of-a-friend without first connecting with the friend) and not the time in-between those connections appearing. We now formalise both types of equivariance:

1. **Permutation equivariance**: Given a permutation $p : [n] \to [n]$ with corresponding matrix $\mathbf{P} \in \mathbb{R}^{n \times n}$, a function $f : \mathbb{R}^{n \times d} \times \mathbb{R}^{n \times n} \to \mathbb{R}^{n \times d}$ is *permutation equivariant* if $f(\mathbf{PX}, \mathbf{PAP}^T) = \mathbf{P}f(\mathbf{X}, \mathbf{A})$ for all permutations $p$ and all graphs with node features $\mathbf{X}$ and adjacency matrix $\mathbf{A}$.

2. **Time-warp equivariance**: Let $X : [0, T] \to Y$ be a continuous path. A *time-warp* is a diffeomorphism (a smooth bijection with smooth inverse) $\tau : [0, T] \to [0, T]$ satisfying $\tau(0) = 0$ and $\tau(T) = T$. A function $f : \chi([0, T], Y) \to \chi([0, T], Y)$ is *time-warp equivariant* if $f(X \circ \tau) = f(X) \circ \tau$ for all time-warps $\tau$ and paths $X$.

For a more formal treatment of group actions and equivariance, we refer the reader to Appendix C.

## 2.2 Neural differential equations for time-series data

Neural Differential Equations (NDEs) ([46, 52, 7, 34]) have emerged as a powerful tool at the intersection of dynamical systems and deep learning. In [8], the authors viewed the layers of a neural network as the time variable of an Ordinary Differential Equation (ODE), introducing the Neural Ordinary Differential Equation (NODE). In NODEs, the time dimension is merely an internal detail of the model, and the trajectories $z$ are entirely determined by the initial condition. To extend the NODE framework to sequential data, [36] introduced *Neural Controlled Differential Equations* (NCDEs) via controlled differential equations (CDEs). For a continuous driving path $X : [0, T] \to \mathbb{R}^{d_x}$ NCDEs learn a latent path $z : [0, T] \to \mathbb{R}^{d_z}$ via the CDE

$$z(0) = \ell_\theta^1(X(0)), \quad z(t) = z(0) + \int_0^t f_\theta(z(s))dX(s) \tag{1}$$

which then returns either a scalar output $y \approx \ell_\theta^2(z(T))$ or an output path $y(t) \approx \ell_\theta^2(z(t))$. Here, $dX(s)$ denotes the Riemann-Stieltjes integral. In practice, we often only have discrete observations $((t_0, X_{t_0}), \ldots, (t_N, X_{t_N}))$ with $t_j \in \mathbb{R}$ and $X_{t_j} \in \mathbb{R}^{d_x}$. To address this, we interpolate the observations into a continuous driving path $X : [t_0, t_N] \to \mathbb{R}^{d_x+1}$ such that $X(t_j) = (t_j, X_{t_j})$ for all $j$. This natively enables NCDEs to process discrete-time sequences or hybrid continuous-discrete dynamics. If the path $X$ is differentiable and has bounded derivative, the Riemann-Stieltjes integral, and hence NCDEs, can be rewritten as the following ordinary integral:

$$z(t) = z(0) + \int_0^t f_\theta(z(s))\frac{\mathrm{d}X(s)}{\mathrm{d}s}\mathrm{d}s \quad \text{for } t \in (0, T]. \tag{2}$$

The choice of interpolation scheme can impact the performance of an NCDE, with [45] providing a discussion on the theoretical properties and practical performance of a range of choices. Additionally, Log-NCDEs [64] extend NCDEs to non-differentiable paths $X$ by leveraging the Log-ODE method to approximate solutions to Equation (1). Empirically, this improves training stability and model performance, especially for long time-series.

## 2.3 Permutation invariant and equivariant linear functions

Graph Neural Networks are typically constructed as permutation equivariant functions by propagating information locally on graphs following a message passing paradigm. An alternative approach, proposed by [44], represents $d$-dimensional graph data on $k$-tuples of nodes as a single matrix $\mathsf{Y} \in \mathbb{R}^{n^k \times d}$. For example, the case $k = 1$ corresponds to node signals and $k = 2$ to signals on edges. In their work, the authors provide a full characterisation of linear maps $L : \mathbb{R}^{n^k \times d} \to \mathbb{R}^{n^k \times d'}$ that are equivariant with respect to permutations of the underlying node set. Remarkably, they show the dimension of the vector space formed by these equivariant maps, denoted by $\mathfrak{E}_{\Sigma_n}(k, d)^{d'}$, depends only on $k$, $d$, and $d'$, and is independent of $n$. In Section 3, we will be interested in the basis of $\mathfrak{E}_{\Sigma_n}(2, 1)^1$, the vector space of linear maps $L : \mathbb{R}^{n^2} \to \mathbb{R}^{n^2}$ between edge-valued data, which has a dimension of $15$. For a list of terms spanning this basis, see Appendix A.

# 3 Permutation equivariant neural graph controlled differential equations

This section brings together Neural CDEs and temporal graph representation learning. We begin by introducing a recent approach proposed in [48], called *Graph Neural Controlled Differential Equations*. Through the lens of Geometric Deep Learning [5], we will identify a key theoretical limitation of this approach and propose an improved solution.

## 3.1 Graph Neural Controlled Differential Equations

The concept of neural controlled differential equations has been extended to graphs by incorporating dynamic adjacency matrices to drive the CDE dynamics [48].

**Graph Neural CDEs**. Let $\zeta_\theta : \mathbb{R}^{n \times n \times 2} \to \mathbb{R}^{n \times d_z}$ and $f_\theta : \mathbb{R}^{n \times d_z} \times \mathbb{R}^{n \times n} \to \mathbb{R}^{(n \times d_z) \times (n \times n \times 2)}$ be two graph neural networks. The *neural controlled differential equation for dynamic graphs* is defined as

$$\mathbf{Z}_t = \mathbf{Z}_{t_0} + \int_{t_0}^{t} f_\theta(\mathbf{Z}_s, \mathbf{A}_s) \mathrm{d}\hat{\mathbf{A}}_s \quad \text{for } t \in (t_0, t_N] \tag{3}$$

where $\mathbf{Z}_{t_0} = \zeta_\theta(\hat{\mathbf{A}}_{t_0})$ is the initial condition, $\hat{\mathbf{A}} : [t_0, t_N] \to \mathbb{R}^{n \times n \times 2}$ such that $\hat{\mathbf{A}}_{t_k}^{i,j} = (t_k, \mathbf{A}_{t_k}^{i,j})$, and the product $f_\theta(\mathbf{Z}_s, \mathbf{A}_s)\mathrm{d}\hat{\mathbf{A}}_s$ is a tensor contraction over $\mathbb{R}^{n \times n \times 2}$. The final prediction $\tilde{\mathbf{Y}}_t \in \mathbb{R}^{n \times d_y}$ is attained by row-wise application of another linear function $\ell_\theta : \mathbb{R}^{d_z} \to \mathbb{R}^{d_y}$, i.e. by slight abuse of notation $\tilde{\mathbf{Y}}_t = \ell_\theta(\mathbf{Z}_t)$.

**Practical Considerations**. Implementing Equation 3 directly is computationally intractable due to the large output dimension of $f_\theta$ and the complexity of the tensor contraction under the integral sign. Hence, [48] proposes the following simplification based on a message passing paradigm:

$$\mathbf{Z}_t = \mathbf{Z}_{t_0} + \int_{t_0}^{t} \mathbf{Z}_s^{(L)} \mathrm{d}s \quad \text{for } t \in (t_0, t_N] \tag{4}$$

where $\mathbf{Z}_s^{(l)} = \sigma(\tilde{\mathbf{A}}_s \mathbf{Z}_s^{(l-1)} \mathbf{W}^{(l-1)})$ for $l \in \{1, \ldots, L\}$, the adjacency matrix and its derivative are fused via $\tilde{\mathbf{A}}_s = \mathbf{W}^{(F)} \begin{bmatrix} \mathbf{A}_s \\ \frac{\mathrm{d}\mathbf{A}_s}{\mathrm{d}s} \end{bmatrix}$ with $\mathbf{W}^{(F)} \in \mathbb{R}^{n \times 2n}$ being a learnable fusion matrix, and $\mathbf{Z}_s^{(0)} = \mathbf{Z}_s$ [3] Importantly, this simplification preserves the multiplicative interaction between the hidden state and control path which are critical for expressivity [13].

## 3.2 Inducing permutation equivariance

As motivated in Section 2, it would be natural to require equivariance with respect to permutation of the node and edge sets for any function on graph data. However, neither the original GN-CDE

---

[3]We note a subsequent version of the GN-CDE introduced an additional fusion matrix, $\tilde{\mathbf{A}}_s = \mathbf{W}^{1,F} \begin{bmatrix} \mathbf{A}_s \\ \frac{\mathrm{d}\mathbf{A}_s}{\mathrm{d}s} \end{bmatrix} \mathbf{W}^{2,F}$ [49]. Our theoretical analysis still is valid, as the fusion remains a linear map.

formulation in Equation 3, nor its approximation in Equation 4 satisfy this property. See Appendix E for a detailed proof.

In the following, we introduce a fully permutation equivariant variant of the GN-CDE framework. By leveraging the characterisation of linear permutation equivariant layers presented in Section 2.3, our goal is to approximate Equation 4 while maintaining equivariance. Since the equivariance breaks in the fusion step, our strategy is to project the fusion operation onto the subspace of linear equivariant functions. Specifically, if we interpret the fusion as the application of linear maps $L_1, L_2 : \mathbb{R}^{n \times n} \to \mathbb{R}^{n \times n}$ to the matrices $\mathbf{A}_s$ and $\frac{\mathrm{d}\mathbf{A}_s}{\mathrm{d}s}$, respectively, we can project $L_1$ and $L_2$ onto $\mathfrak{E}_{\Sigma_n}(2,1)^1$. According to the Projection Theorem (see Appendix B), this yields the best possible approximation. This motivates us to propose the following:

**The model**. Let $L_1, L_2 \in \mathfrak{E}_{\Sigma_n}(2,1)^1$ be two learnable permutation equivariant linear maps, i.e. a weighted combination of the basis terms in Appendix A. Using these, we combine the adjacency and its time derivative as

$$\bar{\mathbf{A}}_s = L_1(\mathbf{A}_s) + L_2\left(\frac{\mathrm{d}\mathbf{A}_s}{\mathrm{d}s}\right). \tag{5}$$

Now, let $\sigma$ be a non-linear activation function and denote by $\mathbf{Z}_s^{(L)}$ the latent representation obtained by an iterative convolution operation of the form $\mathbf{Z}_s^{(l)} = \sigma(\tilde{\mathbf{A}}_s \mathbf{Z}_s^{(l-1)} \mathbf{W}^{(l-1)})$ for $l \in \{1, \dots, L\}$ where the $\mathbf{W}^{(l)}$ are learnable matrices. The *Permutation Equivariant Neural Graph Controlled Differential Equation (PENG-CDE)* then takes the form

$$\mathbf{Z}_t = \mathbf{Z}_{t_0} + \int_{t_0}^{t} \sigma(\bar{\mathbf{A}}_s \mathbf{Z}_s^{(L)} \mathbf{W}^{(L)}) \mathrm{d}s \quad \text{for } t \in (t_0, t_N]. \tag{6}$$

with initial condition $\mathbf{Z}_{t_0} = \zeta_\theta(\hat{\mathbf{A}}_{t_0})$.

**Theoretical Properties**. In the purely linear case (i.e., without applying the non-linearities $\sigma$ in between the layers), one can show that the notion of optimality above, as motivated by the Projection Theorem, is satisfied, as formalised in the following theorem:

**Theorem 3.1.** *In the absence of non-linearities, the PENG-CDE model in Equation 6 is the projection of the model in Equation 4 onto the space of equivariant linear functions.*

*Proof sketch.* Projecting an the flow of an ODE onto a function space is equivalent to projecting its associated vector field. Moreover, note that the vector field in Equation 4 can be decomposed as the composition of a standard convolutional graph neural network, which is inherently equivariant, and the adjacency fusion, which is not. Thus, projecting Equation 4 is equivalent to projecting only the fusion operator. For a detailed proof, see Appendices C and D. $\square$

Although the projection-based notion of optimality holds strictly only in the linear case, by construction, our model satisfies both equivariance constraints outlined in Section 2.1.

**Proposition 3.2.** *The PENG-CDE model in Equation 6 is both permutation equivariant in the spatial domain and time-warp equivariant in the temporal domain.*

*Proof.* See Appendix E. $\square$

### 3.3 Including dynamic node features

So far, our framework only incorporates dynamic adjacency matrices into the latent dynamics, leaving dynamic node features unaddressed. To remedy this, we draw on the approach of [11]. Concretely, suppose node feature snapshots $\{X_{t_k}\}_{k=0}^{N}$ are available, where each $X_{t_k} \in \mathbb{R}^{d_x}$. We interpolate these snapshots to obtain a continuous, differentiable path $\mathbf{X} : [t_0, t_N] \to \mathbb{R}^{d_x+1}$ as in Section 2.2. Next, we choose $\mathbf{W}^{(L)} \in \mathbb{R}^{d_z \times (d_z \times (d_x+1))}$ to set the output dimension of $\sigma\left(\bar{\mathbf{A}}_s \mathbf{Z}_s^{(L)} \mathbf{W}^{(L)}\right)$ to $\mathbb{R}^{n \times (d_z \times (d_x+1))}$, enabling a row-wise (i.e., node-wise) Hadamard multiplication, denoted by $\odot$. The resulting system is

$$\mathbf{Z}_t = \mathbf{Z}_{t_0} + \int_{t_0}^{t} \left\{ \sigma\left(\bar{\mathbf{A}}_s \mathbf{Z}_s^{(L)} \mathbf{W}^{(L)}\right) \odot \frac{\mathrm{d}\mathbf{X}_s}{\mathrm{d}s} \right\} \mathrm{d}s \quad \text{for} \quad t \in (t_0, t_N]. \tag{7}$$

Building on Proposition 3.2, this extension preserves both permutation equivariance in the spatial domain and time-warp equivariance in the temporal domain. Intuitively, this is because a permutation of the node indices reorders both $\sigma\left(\bar{\mathbf{A}}_s \mathbf{Z}_s^{(L)} \mathbf{W}^{(L)}\right)$ and $\frac{\mathrm{d}\mathbf{X}_s}{\mathrm{d}s}$ in the same manner, ensuring that the overall model remains equivariant. A detailed proof is provided in Appendix E. '

# 4 Numerical experiments

In this section, we evaluate the PENG-CDE model on a range of synthetic and real-world tasks. First, we replicate the experiments from [48] and compare them with the non-permutation-equivariant version. Next, we evaluate the model on well-established real-world dynamic graph benchmarks against other common approaches. Finally, we conduct an ablation study to examine the weighting of different basis terms of $\mathfrak{E}_{\Sigma_n}(2,1)^1$ in Appendix 4.3. For supplementary information regarding implementation details, see Appendix G. Appendix H contains a statistical analysis of the results.

## 4.1 Synthetic experiments: heat diffusion and gene regulation

**Task.** We randomly sample initial graphs from four distinct graph distributions (grid, small-world, power-law, and community), each comprising 400 nodes. We then take 120 irregularly sampled time-stamps spanning $T = 0$ to $T = 5$. At 12 time steps, sampled uniformly at random from these snapshots, we randomly add or remove edges following a Bernoulli trial. The final 20 snapshots are allocated for extrapolation validation, while from the remaining 100, a random subset of 20 is used for interpolation validation and the remaining 80 for training. This means the expected number of times the graph topology changes during training and validation phases are 8 and 4, respectively. A batch of four such time series are generated for each of training, validation, and testing. We then simulate node features according to the heat diffusion dynamics governed by Newton's law of cooling and the gene regulatory dynamics governed by the Michaelis–Menten equation. For additional experiments on personal capital and opinion dynamics, see Appendix F.

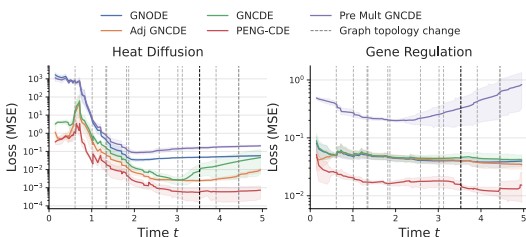

Figure 1: Test losses, plotted against simulation time, for the Graph Neural ODE, three GN-CDE variants, and our proposed Permutation-Equivariant GN-CDE model on the heat diffusion (left) and gene regulation (right) tasks. Dashed vertical lines mark changes in graph topology, while the bold black line indicates the final time point in the training set. Results are reported as means (solid) and ranges (shaded) over a test set with a batch size of four.

**Baselines.** We compare the performance of several models, including state-of-the-art recurrent (DCRNN [40]), interactive (STIDGCN [41]), and attentional (ASTGCN [26]) baselines. For differential equation–based baselines, we first consider a simple model with a constant vector field (Const), that is $\mathbf{Z}_s^{(L)} = \mathbf{b}$ for all $s \in [t_0, t_N]$ in Equation 4. Next, we consider the Graph Neural ODE (GNODE) model [47], in which the adjacency matrix within the vector field is discretised by flooring the time index. The GNODE is governed by the equation:

$$\mathbf{Z}_t = \mathbf{Z}_{t_0} + \int_{t_0}^{t} f_\theta(\mathbf{Z}_s, \mathbf{A}_{\lfloor s \rfloor})\mathrm{d}s \qquad (8)$$

for all $t \in (t_0, t_N]$ where $\mathbf{A}_{\lfloor s \rfloor}$ is the adjacency matrix corresponding to the graph $\mathcal{G}_{t_k}$ with $t_k = \lfloor s \rfloor$. Additionally, we consider several variants of the GN-CDE. Firstly, we include a simplified fusion model, named Adjacency GN-CDE, which employs interpolated adjacency matrices, i.e. $\tilde{\mathbf{A}}_s = \mathbf{A}_s$. We note that for the experiments on these two datasets, the GN-CDE [48] model implemented the fusion by a simple element-wise summation between the adjacency matrix and its derivative (i.e. $\tilde{\mathbf{A}}_s = \mathbf{A}_s + \frac{\mathrm{d}\mathbf{A}_s}{\mathrm{d}s}$), which implicitly achieves permutation equivariance - a property not present in the original formulation. To ensure a fair comparison with a non-equivariant GN-CDE variant, we also consider the Premultiplication Fusion GN-CDE (Pre Mult GN-CDE) model. In this variant, the fusion step is performed via premultiplication by learnable matrices $\mathbf{W}_1$ and $\mathbf{W}_2$, such that $\tilde{\mathbf{A}}_s = \mathbf{W}_1\mathbf{A}_s + \mathbf{W}_2\frac{\mathrm{d}\mathbf{A}_s}{\mathrm{d}s}$. Lastly, we include our proposed Permutation Equivariant Neural Graph CDE (PENG-CDE). Experimental results for the heat diffusion and gene regulation tasks are summarised in Table 1. Moreover, we visu-

alise the test loss over simulation time for models initialised from the community graph distribution in Figure 1.

**Finding I: Equivariance improves performance.** Incorporating permutation equivariance (found in models Adjacency GN-CDE, Original GN-CDE, and PENG-CDE) leads to performance improvements of an order of magnitude over the non-equivariant Pre Mult GN-CDE across all considered tasks and graph distributions.

**Finding II: Enhanced expressivity via 15 basis terms boosts performance.** Integrating all 15 basis terms of linear equivariant maps into the fusion of our model significantly enhances its expressivity. Compared to the original implementation, the PENG-CDE achieves relative MSE improvements ranging from 30.44% to 73.84% on the heat diffusion task and from 39.71% to 67.06% on gene regulation tasks. One plausible explanation is that summing across rows (basis term 3 in Appendix A) corresponds to computing node degrees in an unweighted graph, which facilitates degree normalisation as required in Equation 26.

**Finding III: Additional terms enhance extrapolation behaviour.** As shown in Figure 1, for the heat diffusion task, our PENG-CDE is the only model capable of maintaining constant losses during the extrapolation phase (i.e. over the final 20 snapshots). This suggests that the added equivariant terms contribute to more robust extrapolation, likely due to the mechanisms discussed above.

Table 1: Comparison of GN-CDE variants and baselines on the heat diffusion (top) and gene regulation (bottom) tasks. Mean MSEs with 95% confidence intervals are reported, with the best mean highlighted in **bold**, and all results within the corresponding confidence interval are underlined. The final row in each table reports the relative improvement of PENG-CDE over the original GN-CDE formulation.

| **Heat Diffusion Task (MSE ↓)** | | | | |
|---|---|---|---|---|
| **Model** | **Community** | **Grid** | **Power Law** | **Small World** |
| DCRNN [40] | $0.722 \pm 1.145$ | $70.392 \pm 108.473$ | $1.690 \pm 1.456$ | $84.690 \pm 105.696$ |
| STIDGCN [41] | $0.554 \pm 0.393$ | $4.297 \pm 0.975$ | $0.907 \pm 0.268$ | $1.826 \pm 0.240$ |
| ASTGCN [26] | $2.188 \pm 0.656$ | $15.480 \pm 1.757$ | $3.849 \pm 0.832$ | $8.269 \pm 1.094$ |
| STG-NCDE [11] | $2.091 \pm 0.645$ | $11.989 \pm 1.090$ | $3.518 \pm 1.105$ | $6.902 \pm 1.303$ |
| Constant | $1.936 \pm 0.550$ | $11.155 \pm 0.669$ | $3.147 \pm 0.850$ | $6.286 \pm 1.056$ |
| Graph Neural ODE [47] | $0.237 \pm 0.322$ | $1.001 \pm 0.751$ | $\underline{0.270 \pm 0.310}$ | $\underline{0.311 \pm 0.268}$ |
| Adjacency GN-CDE | $0.208 \pm 0.240$ | $0.691 \pm 0.887$ | $\mathbf{0.258 \pm 0.288}$ | $\underline{0.248 \pm 0.240}$ |
| Pre Mult GN-CDE | $1.968 \pm 0.548$ | $12.262 \pm 1.437$ | $7.440 \pm 4.458$ | $6.829 \pm 0.644$ |
| Original GN-CDE [48] | $0.366 \pm 0.400$ | $1.324 \pm 0.630$ | $\underline{0.417 \pm 0.314}$ | $0.552 \pm 0.470$ |
| PENG-CDE (ours) | $\mathbf{0.096 \pm 0.051}$ | $\mathbf{0.481 \pm 0.195}$ | $\underline{0.290 \pm 0.265}$ | $\mathbf{0.247 \pm 0.215}$ |
| Relative Improvement | 73.84% | 63.70% | 30.44% | 55.20% |

| **Gene Regulation Task (MSE ↓)** | | | | |
|---|---|---|---|---|
| **Model** | **Community** | **Grid** | **Power Law** | **Small World** |
| DCRNN | $159.095 \pm 149.970$ | $28.784 \pm 16.926$ | $77.469 \pm 28.178$ | $27.300 \pm 11.641$ |
| STIDGCN | $14.579 \pm 2.815$ | $0.633 \pm 0.163$ | $4.828 \pm 0.694$ | $0.611 \pm 0.162$ |
| ASTGCN | $13.366 \pm 3.302$ | $0.695 \pm 0.178$ | $4.722 \pm 0.644$ | $0.579 \pm 0.116$ |
| STGNCDE | $88.354 \pm 15.126$ | $8.753 \pm 0.898$ | $20.498 \pm 2.633$ | $6.760 \pm 0.787$ |
| Constant | $36.307 \pm 2.609$ | $1.390 \pm 0.168$ | $7.099 \pm 0.382$ | $0.772 \pm 0.126$ |
| Graph Neural ODE [47] | $8.548 \pm 3.212$ | $\mathbf{0.167 \pm 0.191}$ | $\mathbf{0.372 \pm 0.398}$ | $\underline{0.294 \pm 1.308}$ |
| Adjacency GN-CDE | $8.909 \pm 6.311$ | $1.476 \pm 2.504$ | $\underline{0.498 \pm 0.126}$ | $\underline{0.195 \pm 0.046}$ |
| Pre Mult GN-CDE | $153.084 \pm 149.609$ | $2.553 \pm 0.251$ | $6.978 \pm 3.045$ | $1.591 \pm 0.146$ |
| Original GN-CDE [48] | $10.717 \pm 7.079$ | $0.457 \pm 0.167$ | $0.822 \pm 0.299$ | $\underline{0.323 \pm 0.154}$ |
| PENG-CDE (ours) | $\mathbf{4.566 \pm 2.780}$ | $\underline{0.247 \pm 0.090}$ | $\underline{0.526 \pm 0.220}$ | $\mathbf{0.186 \pm 0.484}$ |
| Relative Improvement | 67.06% | 45.90% | 39.71% | 54.14% |

Table 2: Results of experiments for the `england-covid` and `twitter-tennis` tasks from the Pytorch Geometric Temporal datasets [55]. Mean values and $95\%$ confidence intervals are reported, with the best mean highlighted in **bold** and all results within the confidence interval around the best mean are underlined.

| Model | england-covid MSE ↓ | | twitter-tennis MSE ↓ | |
| --- | --- | --- | --- | --- |
| | Validation | Test | Validation | Test |
| DCRNN [40] | $0.898 \pm 0.126$ | $1.021 \pm 0.203$ | $0.472 \pm 0.136$ | $0.455 \pm 0.087$ |
| ASTGCN [26] | $1.235 \pm 0.139$ | $1.283 \pm 0.257$ | $0.545 \pm 0.128$ | $0.530 \pm 0.078$ |
| STIDGCN [41] | $0.916 \pm 0.111$ | $\underline{0.933 \pm 0.100}$ | $0.524 \pm 0.140$ | $0.495 \pm 0.059$ |
| GNODE [47] | $\mathbf{0.802 \pm 0.184}$ | $\underline{0.945 \pm 0.251}$ | $\underline{0.419 \pm 0.068}$ | $\underline{0.525 \pm 0.043}$ |
| STG-NCDE [11] | $1.484 \pm 0.515$ | $1.778 \pm 1.084$ | $\underline{0.372 \pm 0.108}$ | $\underline{0.453 \pm 0.035}$ |
| GN-CDE [48] | $0.892 \pm 0.143$ | $\underline{0.962 \pm 0.278}$ | $\mathbf{0.369 \pm 0.070}$ | $\underline{0.443 \pm 0.053}$ |
| PENG-CDE (ours) | $\underline{0.836 \pm 0.122}$ | $\mathbf{0.913 \pm 0.200}$ | $\underline{0.391 \pm 0.069}$ | $\mathbf{0.440 \pm 0.053}$ |

**Oversampling and irregularity.** In Figure 2, we compare the performance of the PENG-CDE with DCRNN [40], STIDGCN [41] and ASTGCN [26] on data generated from the graph SIR disease spread model [33]. Its parameters are chosen to produce trajectories both with and without an outbreak, and the task is to classify each trajectory into one of the two categories. To study the effect of oversampling, we simulate each trajectory up to a fixed terminal time $T = 1$ and train and evaluate the models on datasets with an increasing number of observation points (left and middle panels). We also investigate how performance varies with the regularity of the sampling grid: by drawing observation times from a Gamma distribution with controlled shape parameter $k$, we obtain series with different degrees of irregularity and plot test accuracy as irregularity increases (right panel). For implementation details, see Appendix G.3.

**Finding IV: PENG-CDE is robust to oversampling and irregular sampling.** CDE-based models such as PENG-CDE decouple the computational complexity of their forward passes from the number of input observations: the number of vector field evaluations is determined by the ODE solver, not by the sampling rate of the data. This property makes them inherently robust to oversampling. As shown in Figure 2, increasing the sampling frequency negatively impacts both the performance and runtime of

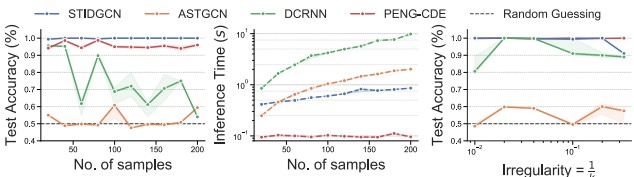

Figure 2: Classification accuracy on the SIR model for STIDGCN, ASTGCN, DCRNN, and our PENG-CDE as a function of the number of observed timesteps (left) and sampling irregularity (right). Inference time with increasing numbers of observations is shown in the middle panel.

the recurrent baseline (DCRNN), while our model remains largely unaffected. Although STIDGCN also maintains performance under oversampling, its computational cost increases with the number of observations, in contrast to the constant runtime of our approach. Likewise, PENG-CDE sustains high performance across all levels of irregularity, while the other models exhibit degraded performance.

## 4.2 Real-world tasks

We evaluate our framework on two types of real-world datasets: snapshot-based and event-based. First, we consider two node regression tasks from PyTorch Geometric Temporal [55] and then two node affinity prediction tasks from the Temporal Graph Benchmark (TGB) [29, 24]. Implementation details are provided in Appendix G.4.

**Snapshot-based datasets.** In this setting, we observe a sequence of dense graphs, one at each timestep. To demonstrate that our modifications enable our model to better capture dynamic graph topologies, we compare it against several differential equation-based models: the Graph Neural ODE [47], the Spatio-Temporal Graph Neural Controlled Differential Equation (STG-NCDE) [11], and the original GN-CDE [48]. Additionally, we include comparisons with recurrent (DCRNN [40]),

attentional (ASTGCN [26]), and interactive (STIDGCN [41]) models. We benchmark performance using the `england-covid` and `twitter-tennis` datasets from PyTorch Geometric Temporal [55] - the only datasets in the library exhibiting dynamic graph topologies. Table 2 reports the mean MSE with 95% confidence intervals on both the validation and test sets over 10 random seeds.

**Finding V: PENG-CDE generalises well on dynamic snapshot datasets.** While not achieving the lowest validation error, PENG-CDE attains the lowest test error on both tasks, indicating better generalisation and reduced overfitting compared to other models.

Table 3: Experimental results for the `tgbn-trade` and `tgbn-genre` node affinity prediction tasks from the Temporal Graph Benchmark datasets [29, 24]. Results marked with † are from [29], those with ‡ are from [68], and those with * are from [59]. Mean values and standard deviations are reported with results within one standard deviation of the best mean highlighted in **bold** (for deterministic and learned models separately).

| Model | trade | genre |
|---|---|---|
| | **NDCG@10 ↑** | |
| Persistent Forecast (L)[†] | **0.855** | 0.357 |
| Moving Avg (L)[†] | 0.823 | **0.509** |
| Moving Avg (M) | 0.777 | 0.472 |
| JODIE[‡] [39] | 0.374±0.09 | 0.350±0.04 |
| TGAT[‡] [15] | 0.375±0.07 | 0.352±0.03 |
| CAWN[‡] [66] | 0.374±0.09 | – |
| TCL[‡] [65] | 0.375±0.09 | 0.354±0.02 |
| GraphMixer[‡] [14] | 0.375±0.11 | 0.352±0.03 |
| DyGFormer[‡] [68] | 0.388±0.64 | 0.365±0.20 |
| DyRep[†] [60] | 0.374±0.001 | 0.351±0.001 |
| TGN[†] [53] | 0.374±0.001 | 0.367±0.058 |
| TGNv2[*] [59] | **0.735**±0.006 | 0.469±0.002 |
| STG-NCDE [10] | 0.618±0.024 | 0.438±0.038 |
| GN-CDE [48] | 0.713±0.026 | 0.460±0.016 |
| PENG-CDE | 0.716±0.029 | **0.523**±0.017 |
| + Source/Target Id | **0.734**±0.024 | – |

**Event-based datasets.** In many applications, a temporal graph is represented as a sequence of events $\mathcal{G} = \{(e_i, t_i, x_i)\}_{i=1}^n$, where each event $e_i$ (e.g., an interaction between nodes $u$ and $v$) occurs at time $t_i$ and is associated with data $x_i \in \mathbb{R}^d$. Since GN-CDEs are not inherently designed for processing individual events, we aggregate events into snapshots. In more detail, given a time window of length $\Delta t$, we partition the overall time span $[t_0, t_n]$ into $m = \lfloor \frac{t_n - t_0}{\Delta t} \rfloor$ non-overlapping intervals. For each interval indexed by $k \in \{0, 1, \ldots, m - 1\}$, we define the snapshot graph $\mathcal{G}_k$ to include all events (or edges) that occur within the time window $[t_0 + k\Delta t, t_0 + (k + 1)\Delta t]$. This then gives us a sequence of snapshots $\{\mathcal{G}_0, \ldots, \mathcal{G}_{m-1}\}$, which we can integrate into the setting above. We evaluate our framework in this event-based setting on the `trade` and `genre` node affinity prediction tasks from the Temporal Graph Benchmark (TGB) [29, 24]. These two tasks were selected from the four available in the benchmark, as the dense adjacency matrices required by Neural ODE-based models make the remaining tasks computationally infeasible within our constraints.

**Baselines.** For the event-based experiments, we first compare against three simple heuristics based on ground-truth labels and messages: 'Persistent Forecast (L)' and 'Moving Average (L)', which operate on labels, and 'Moving Average (M)', which is applied to messages. We then benchmark several learned models, including JODIE [39], TGAT [15], CAWN [66], TCL [65], GraphMixer [14], DyGFormer [68], DyRep [60], TGN [53], and TGNv2 [59]. Additionally, we include STG-NCDE [10], GN-CDE [48][4], and our proposed PENG-CDE. It is a known issue that the heuristics above often outperform learned models on node-affinity prediction tasks [24]; see [59] for a detailed discussion. This motivated TGNv2 to introduce a mechanism that learns node embeddings to distinguish source and target nodes for each interaction, a feature tailored specifically to the node affinity prediction task. To ensure a fair comparison, we incorporate an analogous mechanism into a variant of our model. Further details are provided in Appendix G.5. The experimental results are summarised in Table 3.

**Finding VI: PENG-CDE achieves state-of-the-art performance on TGB.** Without source-target identification, our model is outperformed only by TGNv2 on the `tgbn-trade` task; with source-target identification, it performs on par. Moreover, on `tgbn-genre`, our model significantly outperforms all baselines; notably, it is the only machine-learned approach to surpass all heuristic baselines.

---

[4]Here, the GN-CDE employs the original element-wise additive fusion, $\tilde{\mathbf{A}}_s = \mathbf{A}_s + \frac{d\mathbf{A}_s}{ds}$ [48]. Concurrently to this work, Qin et al. [49] proposed a non-linear element-wise fusion for GN-CDE tailored to the TGBN datasets that achieves comparable performance to the permutation-equivariant fusion of PENG-CDE.

### 4.3 Ablation studies

**Fusion.** To understand how our model fuses the adjacency matrix and its derivative, we analyse the learned weights of the 15 basis terms (Table 4) in the heat diffusion task described in Section 4.1. Both GCN layers assign the largest weights to the identity operations. This is expected, given heat diffusion aggregates over node neighbourhoods. Because the constructed graph is undirected, the transpose operation is equivalent to the identity, allowing the corresponding weight in layer 1 to be disregarded. Layer 1 also

Table 4: Fusion operation weights of the Permutation Equivariant GN-CDE model. Weights with absolute value larger than $0.1$ are in bold.

| Operation | | Layer 1 | | Layer 2 | |
|---|---|---|---|---|---|
| | | $\mathbf{A}_s$ | $d\mathbf{A}_s$ | $\mathbf{A}_s$ | $d\mathbf{A}_s$ |
| Identity | | **1.6129** | **0.9591** | **1.0747** | **0.9100** |
| Transpose | | **0.1026** | −0.0210 | −0.0270 | 0.0535 |
| Diagonalisation | | 0.0297 | −0.0754 | 0.0072 | 0.0317 |
| | rows | **0.7753** | −0.0635 | 0.0358 | −0.0114 |
| Sum rows on | columns | **0.3407** | 0.0015 | −0.0242 | 0.0790 |
| | diagonal | −0.0875 | −0.0222 | −0.0707 | 0.0869 |
| Sum all on | all | **0.4315** | **0.4260** | 0.0039 | 0.0020 |
| | diagonal | −0.0002 | −0.0001 | 0.0362 | 0.0088 |

assigns importance to non-identity components. Large weights are given to the summations of adjacency matrix rows and columns, corresponding to node degree computation, which indicates attention to global graph properties. Similarly, large weights are observed for the total sums of both the adjacency matrix and its derivative, which may correspond to a form of normalisation. Overall, we see that the model utilises the additional expressivity over GN-CDEs.

## 5 Conclusion

In this work, we introduced Permutation Equivariant Neural Graph CDEs - a geometrically grounded approach to temporal graph representation learning that balances parameter efficiency with formal expressivity. In the linear setting, we showed that our model can be derived as a projection of Graph Neural CDEs onto the space of permutation equivariant functions. Through synthetic experiments, we demonstrated that both the imposed equivariance and enhanced expressivity improve performance in both interpolation and extrapolation tasks. Additionally, our model retains the strengths of Neural CDEs in handling oversampling and irregular time intervals. On the TGB-genre dataset, our model achieves a new state-of-the-art – becoming the first learned method to surpass a moving average baseline.

### 5.1 Limitations and future work

Like the original GN-CDE, PENG-CDEs have quadratic memory complexity in the number of nodes due to the need to store dense adjacency matrices, which restricts scalability to large, sparse graphs. This constraint arises primarily from JAX's limited support for sparse matrix operations. As shown in Table 11 in Appendix I, once memory allocation constraints are met, PENG-CDE scales much more favourably in runtime compared to GN-CDE – demonstrating its potential for efficient modelling at larger scales.

Future work includes addressing this scalability bottleneck and investigating the impact of more tailored hyperparameter configurations, such as the choice of interpolation schemes, ODE solvers, and vector field architectures. In addition, advanced solvers like Log-NCDE [64], originally developed for Euclidean paths, could be adapted to the graph setting. While we chose dynamic adjacency matrices as the control signal for the CDE, this is just one possible vectorisation of graph structure. Prior work [25] suggests that adjacency matrices are often suboptimal for representing graph properties. Identifying more expressive or task-aligned graph representations could further improve performance and remains an exciting research direction. Finally, while Neural CDEs are known to be universal approximators on Euclidean domains [36], and recent work has also provided generalisation bounds [3], no such theoretical guarantees currently exist for graph-based Neural CDEs. Establishing these would be a valuable theoretical contribution to the growing field of temporal graph representation learning.

## Acknowledgements

This work is supported by the Helmholtz Association Initiative and Networking Fund on the HAICORE@KIT partition. Benjamin Walker is funded by the Hong Kong Innovation and Technology Commission (InnoHK Project CIMDA).

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

## A Permuation equivariant linear basis terms

Maron et al. [44] derive the 15 basis elements of all linear permutation equivariant maps $L : \mathbb{R}^{n^2} \to \mathbb{R}^{n^2}$ on edge-valued signals $\mathbf{A} \in \mathbb{R}^{n^2 \times 1}$. They show the following 15 maps span this space:

We denote by $\mathbf{1} \in \mathbb{R}^n$ the vector whose entries are all equal to one, and by diag the operator that either places a vector on the diagonal of a matrix or extracts the diagonal of a matrix, depending on context. With this notation, the basis elements of $\mathfrak{E}_{\Sigma_n}(2,1)^1$ can be listed explicitly as follows:

1. The identity and transpose operations: $L(\mathbf{A}) = \mathbf{A}, L(\mathbf{A}) = \mathbf{A}^T$,

2. Elimination of non-diagonal elements: $L(\mathbf{A}) = \text{diag}(\text{diag}(\mathbf{A}))$,

3. Sum of rows replicated on rows/columns/diagonal: $L(\mathbf{A}) = \mathbf{1}\mathbf{1}^T \mathbf{A}, L(\mathbf{A}) = \mathbf{1}(\mathbf{A}\mathbf{1})^T, L(\mathbf{A}) = \text{diag}(\mathbf{A}\mathbf{1})$,

4. Sum of columns replicated on rows/columns/diagonal: $L(\mathbf{A}) = \mathbf{A}^T \mathbf{1}\mathbf{1}^T, L(\mathbf{A}) = \mathbf{1}(\mathbf{A}^T\mathbf{1})^T, L(\mathbf{A}) = \text{diag}(\mathbf{A}^T\mathbf{1})$,

5. Sum of all matrix entries replicated on all entries/diagonal: $L(\mathbf{A}) = \mathbf{1}^T\mathbf{A}\mathbf{1} \cdot \mathbf{1}\mathbf{1}^T, L(\mathbf{A}) = \mathbf{1}^T\mathbf{A}\mathbf{1} \cdot \text{diag}(\mathbf{1})$,

6. Sum of all diagonal entries replicated on all entries/diagonal: $L(\mathbf{A}) = \mathbf{1}^T\text{diag}(\mathbf{A}) \cdot \mathbf{1}\mathbf{1}^T, L(\mathbf{A}) = \mathbf{1}^T\text{diag}(\mathbf{A}) \cdot \text{diag}(\mathbf{1})$,

7. Diagonal elements replicated on rows/columns: $L(\mathbf{A}) = \text{diag}(\mathbf{A})\mathbf{1}^T, L(\mathbf{A}) = \mathbf{1}\text{diag}(\mathbf{A})^T$,

## B Projection theorem

**Definition B.1** (Projection). *Let $\mathcal{H}$ be a Hilbert space with inner product $(\cdot, \cdot)$. A linear map $p : \mathcal{H} \to \mathcal{H}$ is called a* projection *if $p \circ p = p$. A projection is called* orthogonal *if for all $x \in \mathcal{H}$ it holds that for all $y \in \mathcal{H}$, $p(y) = 0$ implies $(p(x), y) = 0$.*

Intuitively, this is saying that applying a projection operator twice yields the same result as applying it once, which captures the idea of "selecting" or "filtering out" a specific component of a vector. The additional condition for an orthogonal projection ensures that the difference between any vector and its projected counterpart is perpendicular to the subspace, embodying the concept of the closest approximation in the geometry of the space.

**Theorem B.2** (Projection Theorem, e.g. [1], Chapter 1). *Let $\mathcal{H}$ be a Hilbert space and $U$ a closed linear subspace. Then every $x \in \mathcal{H}$ can uniquely be written as $x = u + u^*$ where $u \in U$, $u^* \in U^\perp := \{y \in \mathcal{H} : (y, u) = 0 \; \forall u \in U\}$. Moreover, the map $P_U : \mathcal{H} \to \mathcal{H}$ defined as $P_U(x) = u$ is an orthogonal projection and satisfies*

$$\|P_U(x) - x\| = \inf_{u' \in U} \|u' - x\|. \tag{9}$$

*for all $x \in \mathcal{H}$.*

The Projection Theorem asserts that every vector in a Hilbert space can be uniquely decomposed into a sum of two parts: one lying in a closed subspace and the other in its orthogonal complement, thus clarifying the geometric structure of the space. Moreover, it guarantees the projection onto a subspace is the unique map that assigns to each element $x \in \mathcal{H}$ the closest element from that subspace, thereby establishing a rigorous method for optimal approximation in terms of the inner product.

## C General theory of equivariance

In the following, we formalise notions of how symmetries can "act" on data and how we want to define functions that respect these symmetries.

### C.1 Equivariance of static functions

**Definition C.1** (Group Action). *Let $G$ be a group and $X$ a set. A* group action *of $G$ on $X$ is a function $\cdot : G \times X \to X$ satisfying:*

1. **Identity:** *for all $x \in X$, $e \cdot x = x$, where $e$ is the identity element of $G$,*

2. **Compatibility:** *for all $g, h \in G$ and $x \in X$, $(gh) \cdot x = g \cdot (h \cdot x)$.*

This definition formalises the idea that the elements of the group $G$ can act on the set $X$ in a way that reflects the group's structure. It essentially captures the concept of symmetry by showing how each group element transforms or "moves" elements in $X$.

**Definition C.2** (Group Representation). *Let $G$ be a group and $V$ a vector space. A* group representation *of $G$ on $V$ is a homomorphism $\rho : G \to \mathrm{GL}(V)$ such that:*

- **Identity:** *$\rho(e) = I_V$, where $e$ is the identity element of $G$ and $I_V$ is the identity map on $V$,*

- **Compatibility:** *For all $g_1, g_2 \in G$, $\rho(g_1 g_2) = \rho(g_1)\rho(g_2)$.*

This definition realises abstract group elements as concrete, invertible linear transformations on $V$, thereby representing the group's symmetry in a linear framework. It provides a way to analyse the structure of $G$ by studying how it acts on a vector space.

**Definition C.3** (Equivariance). *Let $\rho : G \to \mathrm{GL}(V)$ and $\sigma : G \to \mathrm{GL}(W)$ be representations of a group $G$ on vector spaces $V$ and $W$, respectively. A function $f : V \to W$ is said to be $G$-equivariant if for all $g \in G$ and $v \in V$,*

$$f\big(\rho(g)(v)\big) = \sigma(g)\big(f(v)\big).$$

This definition states that applying the group action before or after the function $f$ produces the same outcome. Equivariance ensures that $f$ preserves the symmetry structure imposed by $G$, making it a natural and symmetry-respecting mapping between the spaces.

## C.2 Haar measure

**Definition C.4** (Haar measure). *Let $G$ be a compact Hausdorff topological group. There exists a unique regular Borel measure $\mu$ on $G$ invariant under both left and right translations:*

$$\mu(gE) = \mu(Eg) = \mu(E) \quad \text{for all } g \in G, \ E \subseteq G \text{ Borel}$$

*with total mass $1$, i.e. $\mu(G) = 1$. We call this measure the Haar measure.*

**Remark C.5.** *If $G$ is finite of order $|G|$, the Haar measure reduces to the uniform distribution:*

$$\mu(\{g\}) = \frac{1}{|G|} \quad \forall g \in G, \qquad \int_G f(g)\, d\mu(g) = \frac{1}{|G|} \sum_{g \in G} f(g).$$

*In this discrete (hence compact) setting, Haar integration is exactly averaging over group elements.*

## C.3 Projection of non-equivariant functions

Given Hilbert spaces $V$ and $W$, the set of functions $\{f : V \to W\}$ form a vector space under pointwise scalar multiplication and addition. Given group representations $\rho$ and $\sigma$ of a group $G$ over $V$ and $W$, respectively, we can consider the subspace of functions that are equivariant with respect to these representations. The following lemma and corollary show what the projection onto this subspace looks like.

**Lemma C.6.** *Let $V$ and $W$ be finite-dimensional vector spaces with an action of a finite group $G$ with representations $\rho : G \to \mathrm{GL}(V)$ and $\sigma : G \to \mathrm{GL}(W)$ respectively. For any linear map $T : V \to W$, define the averaged map by*

$$T_{\mathrm{avg}}(v) = \int_G \sigma(g)^{-1} T\big(\rho(g)v\big)\, \mathrm{d}\,\mu(g).$$

*Then $T_{\mathrm{avg}}$ is $G$-equivariant, and moreover, $T$ is $G$-equivariant if and only if $T = T_{\mathrm{avg}}$.*

*Proof.* To show that $T_{\mathrm{avg}}$ is $G$-equivariant, take any $h \in G$ and $v \in V$. Then

$$T_{\mathrm{avg}}(\rho(h)v) = \int_G \sigma(g)^{-1} T\big(\rho(g)\rho(h)v\big)\, \mathrm{d}\,\mu(g)$$

$$= \int_G \sigma(g)^{-1} T\big(\rho(gh)v\big)\, \mathrm{d}\,\mu(g).$$

By substituting $g' = gh$ (so that $g = g'h^{-1}$), we obtain

$$T_{\text{avg}}(\rho(h)v) = \int_G \sigma(g'h^{-1})^{-1}T\big(\rho(g')v\big)\,\mathrm{d}\,\mu(g')$$
$$= \int_G \sigma(h)\sigma(g')^{-1}T\big(\rho(g')v\big)\,\mathrm{d}\,\mu(g')$$
$$= \sigma(h)T_{\text{avg}}(v),$$

where we used $\sigma(g'h^{-1})^{-1} = \sigma(h)\sigma(g')^{-1}$ because $\sigma$ is a homomorphism. Thus, $T_{\text{avg}}$ is $G$-equivariant.

Next, if $T$ is already $G$-equivariant, then for every $g \in G$,

$$\sigma(g)^{-1}T(\rho(g)v) = T(v),$$

so that

$$T_{\text{avg}}(v) = \int_G T(v) = T(v).$$

$\square$

**Corollary C.7.** *In the setting of the previous lemma, the map* $()_{\text{avg}} : \text{Hom}(V, W) \to \text{Hom}(V, W)$ *defined by* $T \mapsto T_{\text{avg}}$ *is the projection map onto the subspace of $G$-equivariant linear maps.*

*Proof.* In the previous proof we saw that $()_{\text{avg}}$ acts as the identity on the subspace of $G$-equivariant linear maps. Since applying $()_{\text{avg}}$ to any linear map $T$ yields a $G$-equivariant map $T_{\text{avg}}$, $()_{\text{avg}}$ is a projection onto the subspace of $G$-equivariant linear maps. $\square$

Next, we study what form the projection of the composition of a linear equivariant function with another function takes.

**Proposition C.8.** *Let $V, X, W$ be finite-dimensional vector spaces with an action of a finite group $G$ given by representations $\rho_V : G \to \text{GL}(V), \rho_X : G \to \text{GL}(X)$ and $\rho_W : G \to \text{GL}(W)$ respectively. Denote the projection of $\text{Hom}(V, W)$ onto the space of of $G$-equivariant functions with respect to the representations $\rho_V$ and $\rho_W$ by $P_{V,W}$ and similar for the other combinations. Now suppose we have functions $T : V \to W, R : V \to X, S : X \to W$ such that*

- $T = S \circ R$,

- *$S$ is $G$-equivariant and linear.*

*We then have $P_{V,W}(T) = S \circ P_{V,X}(R)$.*

*Proof.* Since $P_{V,W}$ projects onto an equivariant subspace, by the above we know that

$$(P_{V,W}T)(v) = \int_G \rho_W(g)^{-1}T(\rho_V(g)v)\,d\mu(g)$$

for all $v \in V$. Now by definition and the linearity of $S$,

$$(P_{V,W}T)(v) = \int_G \rho_W(g)^{-1}(S \circ R)(\rho_V(g)v)\,d\mu(g)$$
$$= \int_G \rho_W(g)^{-1}S(R(\rho_V(g)v))\,d\mu(g)$$
$$= \int_G S(\rho_X(g)^{-1}R(\rho_V(g)v))\,d\mu(g) \qquad \text{(equivariance of } S)$$
$$= S\left(\int_G \rho_W(g)^{-1}R(\rho_V(g)v)\,d\mu(g)\right) \qquad \text{(linearity of } S)$$
$$= S\left((P_{V,X}R)(v)\right)$$

since $S\big(\rho_V(g)^{-1}R(\rho_V(g)v)\big) = \rho_W(g)^{-1}S(R(\rho_V(g)v))$ by the equivariance of $S$. This concludes the proof. $\square$

## C.4 Equivariance in Neural ODEs

**Definition C.9.** *Let $f : X \times I \to X$ be a smooth function, where $I \subset \mathbb{R}$ is an interval containing $0$ and $X$ is some Banach space. For each $x \in X$, consider the initial value problem*

$$\frac{dz}{dt} = f(z,t), \quad z(0) = x. \tag{10}$$

*The* flow *of the ODE is the mapping* $\Phi : I \times X \to X, \quad (t,x) \mapsto \Phi_t(x)$*, which assigns to each pair $(t,x)$ the solution to the IVP in equation 10 with initial condition $x$ evaluated at time $t$.*

**Proposition C.10.** *Let $\Phi_t(x)$ be the flow of an ODE with vector field $f : X \times I \to X$ and let $P$ be the projection operator of the space of functions $\{g : X \to X\}$ onto the space $G$-equivariant maps with respect to a group representation $\rho$ of $G$ on $X$. Then the projected flow $P(\Phi_t)$ and the flow $\widetilde{\Phi}_t$ of the projected vector field $P(f) : X \times I \to X$ agree for all times $t \in I$, i.e. $P(\Phi_t) = \widetilde{\Phi}_t$.*

*Proof.* First, we note that by Corollary C.7, $P$ can be realised by

$$P(f)(x,t) = \int_G \rho(g)^{-1} f(\rho(g)x, t) \, \mathrm{d}\,\mu(g),$$

$$P(\Phi_t)(x) = \int_G \rho(g)^{-1} \Phi_t(\rho(g)x) \, \mathrm{d}\,\mu(g).$$

Now, by the uniqueness theory of ODEs (see for example Theorem 6 in [43]), it suffices to show that $P(\Phi_t)$ satisfies the same initial condition and ODE as $\widetilde{\Phi}_t$. To that end, we calculate

$$\frac{d}{dt}\big(P\Phi_t\big)(x) = \int_G \rho(g)^{-1} \frac{d}{dt} \Phi_t(\rho(g)x) \, \mathrm{d}\,\mu(g)$$

$$= \int_G \rho(g)^{-1} f(\rho(g)x, t) \, \mathrm{d}\,\mu(g)$$

$$= P(f)(x,t)$$

and

$$\big(P\Phi_0\big)(x) = \int_G \rho(g)^{-1} \Phi_0(\rho(g)x) \, \mathrm{d}\,\mu(g)$$

$$= \int_G \rho(g)^{-1} \rho(g)x \, \mathrm{d}\,\mu(g)$$

$$= x.$$

This finishes the proof. $\qquad\qquad\square$

This theorem shows that, under the stated conditions, if one projects the full flow $\Phi_t$ using $P$, then this projected flow is exactly the same as the flow obtained by integrating the projected vector field $P(f)$.

# D  Application to Graph Neural CDE

**Definition D.1** (Representation of the Permutation Group). *We define a* representation *of the permutation group $S_n$ on the space of node features $\mathbb{R}^{n \times d}$ for each $\sigma \in S_n$ by $\rho(\sigma)(\boldsymbol{X}) = \boldsymbol{P}_\sigma \boldsymbol{X}$ where $\boldsymbol{P}_\sigma$ is the permutation matrix associated with $\sigma$. Similarly, we define a representation on the space of node features $\mathbb{R}^{n \times n}$ by $\pi(\sigma)(\boldsymbol{X}) = \boldsymbol{P}_\sigma \boldsymbol{X} \boldsymbol{P}_\sigma^T$.*

The pre-multiplication by $\mathbf{P}_\sigma$ permutes the rows of a matrix, and the post-multiplication permutes the columns.

*Proof of Theorem 3.1.* Firstly, by Proposition C.10, the projection of any Graph Neural ODE onto the subspace of permutation equivariant functions is the Neural ODE where the vector field is the projection of the Graph Neural ODE. Hence, it suffices to consider the projection of the integrand in Equation 4.

We can decompose this integrand as the composition of the following two functions:

$$R(\mathbf{Z}, \mathbf{A}, \frac{d\mathbf{A}}{ds}) = \left( \mathbf{Z}, \mathbf{W}^{(DR)} \begin{bmatrix} \mathbf{A} \\ \frac{d\mathbf{A}}{ds} \end{bmatrix} \right)$$

$$S(\mathbf{Z}, \tilde{\mathbf{A}}) = \tilde{\mathbf{A}} \mathbf{Z}^{(L)} \mathbf{W}^{(L)}$$

where $\mathbf{Z}^{(L)}$ are latent representations obtained by an iterative convolutional process of the form $\mathbf{Z}^{(l)} = \tilde{\mathbf{A}} \mathbf{Z}^{(l-1)} \mathbf{W}^{(l-1)}$ for $l \in \{1, \ldots, L\}$ where the $\mathbf{W}^{(l)}$ are learnable matrices.

The function $S$ is a standard convolutional GNN and hence both permutation equivariant and linear in $\mathbf{Z}$. Hence, we can apply Proposition C.8 and realise we only need to consider the projection of $R$.

But since $W^{(DR)}$ is learnable and in Section 2.3 we characterised the space of linear equivariant functions on the space of adjacency matrices, instead of projecting onto the space of equivariant functions, we may directly parameterise an element of this space as a linear combination of the basis elements. This concludes the proof. $\qquad\square$

# E   Proofs

*Proof the GN-CDE (Equation 3) is not permutation equivariant.* For notational simplicity, we flatten $f_\theta(\mathbf{Z}_s, \mathbf{A}_s)$ and the control $d\hat{\mathbf{A}}_s$ into tensors of ranks 2 and 1, respectively:

$$F_\theta(\mathbf{Z}_s, \mathbf{A}_s) = \text{vec}\left[ f_\theta(\mathbf{Z}_s, \mathbf{A}_s) \right] \in \mathbb{R}^{nd_z \times 2n^2}, \tag{11}$$

$$\mathbf{B} = \text{vec}\left[ d\hat{\mathbf{A}}_s \right] \in \mathbb{R}^{2n^2}. \tag{12}$$

Under a node permutation $\sigma \in S_n$, with permutation matrix $\mathbf{P}_\sigma$, these transform as

$$F_\theta(\mathbf{Z}_s, \mathbf{A}_s) \mapsto (\mathbf{P}_\sigma \otimes \mathbf{I}_{d_z}) F_\theta(\mathbf{Z}_s, \mathbf{A}_s), \tag{13}$$

$$\mathbf{B} \mapsto (\mathbf{P}_\sigma \otimes \mathbf{P}_\sigma \otimes \mathbf{I}_2) \mathbf{B}. \tag{14}$$

Requiring that the contraction

$$T_\theta(\mathbf{Z}_s, \mathbf{A}_s) = F_\theta(\mathbf{Z}_s, \mathbf{A}_s) \mathbf{B}$$

be permutation-equivariant then gives

$$T_\theta(\mathbf{P}_\sigma \mathbf{Z}_s, \mathbf{P}_\sigma \mathbf{A}_s \mathbf{P}_\sigma^\top) = (\mathbf{P}_\sigma \otimes \mathbf{I}_{d_z}) T_\theta(\mathbf{Z}_s, \mathbf{A}_s) (\mathbf{P}_\sigma^\top \otimes \mathbf{P}_\sigma^\top \otimes \mathbf{I}_2) \quad \forall \sigma \in S_n.$$

No general architecture for $f_\theta$ (for example, a standard multilayer perceptron) can satisfy this constraint unless it is explicitly parametrised to output elements in the space of permutation-equivariant linear maps. $\qquad\square$

*Proof the GN-CDE fusion operation (Equation 4) is not permutation equivariant.* Firstly we note that by writing $\mathbf{W}^{(F)} = \left[ \mathbf{W}_1^{(F)} \; \mathbf{W}_2^{(F)} \right]$ for $\mathbf{W}_1^{(F)}, \mathbf{W}_2^{(F)} \in \mathbb{R}^{n \times n}$, the fusion takes the form

$$\tilde{\mathbf{A}}_s = \mathbf{W}^{(F)} \begin{bmatrix} \mathbf{A}_s \\ \frac{d\mathbf{A}_s}{ds} \end{bmatrix} = \left[ \mathbf{W}_1^{(F)} \; \mathbf{W}_2^{(F)} \right] \begin{bmatrix} \mathbf{A}_s \\ \frac{d\mathbf{A}_s}{ds} \end{bmatrix} = \mathbf{W}_1^{(F)} \mathbf{A}_s + \mathbf{W}_2^{(F)} \frac{d\mathbf{A}_s}{ds}. \tag{15}$$

Thus, the fusion operation can be viewed as pre-multiplying each of $\mathbf{A}_s$ and $\frac{d\mathbf{A}_s}{ds}$ by a learnable weight matrix and then summing the result. However, letting $\mathbf{P}$ be an arbitrary permutation matrix, we have

$$\mathbf{P}\tilde{\mathbf{A}}_s\mathbf{P}^T = \mathbf{P}\mathbf{W}_1^{(F)}\mathbf{A}_s\mathbf{P}^T + \mathbf{P}\mathbf{W}_2^{(F)}\frac{d\mathbf{A}_s}{ds}\mathbf{P}^T \tag{16a}$$

$$\neq \mathbf{W}_1^{(F)}\mathbf{P}\mathbf{A}_s\mathbf{P}^T + \mathbf{W}_1^{(F)}\mathbf{P}\frac{d\mathbf{A}_s}{ds}\mathbf{P}^T \tag{16b}$$

in general, since $\mathbf{W}_1^{(F)}$ and $\mathbf{W}_2^{(F)}$ do not necessarily commute with $\mathbf{P}$. $\qquad\square$

*Proof of Proposition 3.2.* First we show permutation equivariance which follows immediately from the construction of the linear fusion. For let $\mathbf{P}$ be an arbitrary permutation matrix. Then

$$L_1(\mathbf{P}\mathbf{A}_s\mathbf{P}^T) + L_2\left(\mathbf{P}\frac{\mathrm{d}\mathbf{A}_s}{\mathrm{d}s}\mathbf{P}^T\right) = \mathbf{P}L_1(\mathbf{A}_s)\mathbf{P}^T + \mathbf{P}L_2\left(\frac{\mathrm{d}\mathbf{A}_s}{\mathrm{d}s}\right)\mathbf{P}^T = \mathbf{P}\bar{\mathbf{A}}_s\mathbf{P}^T. \qquad (17)$$

Since $\zeta_\theta$ is implemented as a GNN, $\zeta_\theta(\mathbf{P}\mathbf{X}_{t_0}, \mathbf{P}\mathbf{A}_{t_0}\mathbf{P}^T) = \mathbf{P}\zeta_\theta(\mathbf{X}_{t_0}, \mathbf{A}_{t_0}) = \mathbf{P}\mathbf{Z}_{t_0}$ and so overall for all $t \in (t_0, t_N]$ we have

$$\mathbf{P}\mathbf{Z}_{t_0} + \int_{t_0}^t \sigma(\mathbf{P}\bar{\mathbf{A}}_s\mathbf{P}^T\mathbf{P}\mathbf{Z}_s^{(L)}\mathbf{W}^{(L)})\mathrm{d}s = \mathbf{P}\mathbf{Z}_{t_0} + \int_{t_0}^t \mathbf{P}\sigma(\bar{\mathbf{A}}_s\mathbf{Z}_s^{(L)}\mathbf{W}^{(L)})\mathrm{d}s \qquad (18a)$$

$$= \mathbf{P}\mathbf{Z}_{t_0} + \mathbf{P}\int_{t_0}^t \sigma(\bar{\mathbf{A}}_s\mathbf{Z}_s^{(L)}\mathbf{W}^{(L)})\mathrm{d}s \qquad (18b)$$

$$= \mathbf{P}\left(\mathbf{Z}_{t_0} + \int_{t_0}^t \sigma(\bar{\mathbf{A}}_s\mathbf{Z}_s^{(L)}\mathbf{W}^{(L)})\mathrm{d}s\right) \qquad (18c)$$

$$= \mathbf{P}\mathbf{Z}_t \qquad (18d)$$

using the orthogonality of $\mathbf{P}$ and linearity of the integral.

To show time-warp equivariance, suppose a latent path $\mathbf{Z} : [0, T] \to \mathbb{R}^{n \times d_z}$ satisfies

$$\mathbf{Z}(0) = \mathbf{Z}_0, \qquad \frac{\mathrm{d}\mathbf{Z}}{\mathrm{d}t}(t) = f_\theta(\mathbf{Z}(t), \mathbf{A}(t))\frac{\mathrm{d}\hat{\mathbf{A}}}{\mathrm{d}t}(t). \qquad (19)$$

for some dynamic graph data $\mathbf{A} : [0, T] \to \mathbb{R}^{n \times n}$. Then the warped path $\hat{\mathbf{Z}} := \mathbf{Z} \circ \tau$ satisfies

$$\hat{\mathbf{Z}}(0) = (\mathbf{Z} \circ \tau)(0) = \mathbf{Z}(0) = \mathbf{Z}_0. \qquad (20)$$

and by a simple application of the chain rule we get

$$\frac{\mathrm{d}\hat{\mathbf{Z}}}{\mathrm{d}t}(t) = \frac{\mathrm{d}}{\mathrm{d}t}(\mathbf{Z} \circ \tau)(t) = \frac{\mathrm{d}\mathbf{Z}}{\mathrm{d}t}(\tau(t))\tau'(t) \qquad (21a)$$

$$= f_\theta(\mathbf{Z}(\tau(t)), \mathbf{A}(\tau(t)))\frac{\mathrm{d}\hat{\mathbf{A}}}{\mathrm{d}t}(\tau(t))\tau'(t) \qquad (21b)$$

$$= f_\theta(\hat{\mathbf{Z}}(t), \mathbf{A}(\tau(t)))\frac{\mathrm{d}}{\mathrm{d}t}(\hat{\mathbf{A}} \circ \tau)(t) \qquad (21c)$$

which is what we wanted. $\qquad\square$

*Proof of equivariance in Section 3.3.* First we show that for general $\mathbf{A} \in \mathbb{R}^{n \times m \times d}$ and $\mathbf{B} \in \mathbb{R}^{n \times d}$ we have $\mathbf{P}(\mathbf{A} \odot \mathbf{B}) = \mathbf{P}\mathbf{A} \odot \mathbf{P}\mathbf{B}$. In Einstein notation we have

$$\{\mathbf{P}(\mathbf{A} \odot \mathbf{B})\}_{ij} = \mathbf{P}_{ik}(\mathbf{A} \odot \mathbf{B})_{kj} \qquad (22a)$$

$$= \sum_k \mathbf{P}_{ik}\mathbf{A}_{kjl}\mathbf{B}_{kl}. \qquad (22b)$$

At the same time

$$(\mathbf{P}\mathbf{A} \odot \mathbf{P}\mathbf{B})_{ij} = (\mathbf{P}\mathbf{A})_{ijl}(\mathbf{P}\mathbf{B})_{il} \qquad (23a)$$

$$= \mathbf{P}_{ik}\mathbf{A}_{kjl}\mathbf{P}_{ik'}\mathbf{B}_{k'l} \qquad (23b)$$

$$= \sum_k \mathbf{P}_{ik}\mathbf{A}_{kjl}\mathbf{B}_{kl} \qquad (23c)$$

$$= \{\mathbf{P}(\mathbf{A} \odot \mathbf{B})\}_{ij} \qquad (23d)$$

because $\mathbf{P}_{ik}, \mathbf{P}_{ik'} \in \{0, 1\}$, so the only terms contributing to the sum are those where $\mathbf{P}_{ik} = \mathbf{P}_{ik'}$ which happens if and only if $k = k'$ as $\mathbf{P}$ is a permutation matrix.

Hence, we see that the vector field in equation 7 is permutation equivariant. By the above, we can conclude that the entire model is equivariant. $\qquad\square$

## F  Additional Experiments

Here, we present the results of the additional experiments on personal capital and opinion dynamics.

Table 5: Additional comparison of GN-CDE variants and baselines on the wealth (top) and opinion dynamics (bottom) tasks. Mean MSEs with $95\%$ confidence intervals are reported, with the best mean highlighted in **bold**, and all results within the corresponding confidence interval are underlined. The final row in each table reports the relative improvement of PENG-CDE over the original GN-CDE formulation.

| Wealth Dynamics (MSE ↓) | | | | |
|---|---|---|---|---|
| **Model** | **Community** | **Grid** | **Power Law** | **Small World** |
| DCRNN [40] | $2.639 \pm 1.981$ | $22.619 \pm 17.115$ | $113.564 \pm 236.126$ | $200.486 \pm 280.387$ |
| STIDGCN [41] | $1.862 \pm 0.947$ | $7.296 \pm 2.547$ | $2.124 \pm 1.012$ | $3.133 \pm 1.286$ |
| ASTGCN [26] | $3.431 \pm 1.224$ | $19.694 \pm 2.986$ | $5.153 \pm 1.184$ | $9.246 \pm 2.204$ |
| STG-NCDE | $36.087 \pm 2.454$ | $44.677 \pm 3.755$ | $37.148 \pm 2.742$ | $41.046 \pm 2.205$ |
| Constant | $2.504 \pm 0.473$ | $14.157 \pm 1.746$ | $3.682 \pm 0.873$ | $7.445 \pm 1.200$ |
| Graph Neural ODE [47] | $0.878 \pm 0.388$ | $\mathbf{1.898 \pm 1.342}$ | $\underline{0.950 \pm 0.660}$ | $\underline{0.940 \pm 0.185}$ |
| Adjacency GN-CDE | $0.904 \pm 0.493$ | $\underline{2.193 \pm 1.721}$ | $\underline{1.075 \pm 0.847}$ | $\underline{1.297 \pm 0.765}$ |
| Pre Mult GN-CDE | $7.740 \pm 1.122$ | $24.522 \pm 6.594$ | $39.715 \pm 32.698$ | $16.568 \pm 6.902$ |
| Original GN-CDE | $1.577 \pm 0.781$ | $6.582 \pm 3.273$ | $2.406 \pm 0.965$ | $3.076 \pm 1.193$ |
| PENG-CDE (ours) | $\mathbf{0.522 \pm 0.316}$ | $4.273 \pm 3.818$ | $\mathbf{0.863 \pm 0.330}$ | $\mathbf{0.813 \pm 0.627}$ |
| Relative Improvement (%) | $66.89\%$ | $35.09\%$ | $64.13\%$ | $73.57\%$ |

| Opinion Dynamics (MSE ↓) | | | | |
|---|---|---|---|---|
| **Model** | **Community** | **Grid** | **Power Law** | **Small World** |
| DCRNN [40] | $20.659 \pm 32.149$ | $92.165 \pm 132.026$ | $91.687 \pm 81.638$ | $173.554 \pm 183.913$ |
| STIDGCN [41] | $2.434 \pm 1.562$ | $6.977 \pm 2.451$ | $2.724 \pm 1.530$ | $3.933 \pm 1.567$ |
| ASTGCN [26] | $22.001 \pm 1.519$ | $34.855 \pm 2.370$ | $22.517 \pm 1.490$ | $27.439 \pm 1.554$ |
| STG-NCDE | $57.495 \pm 9.245$ | $85.405 \pm 6.479$ | $59.446 \pm 9.294$ | $68.092 \pm 8.988$ |
| Constant | $22.989 \pm 1.372$ | $33.124 \pm 1.275$ | $23.775 \pm 1.480$ | $27.239 \pm 1.929$ |
| Graph Neural ODE [47] | $\underline{0.665 \pm 0.620}$ | $\underline{2.338 \pm 1.277}$ | $3.073 \pm 6.114$ | $\mathbf{0.779 \pm 0.417}$ |
| Adjacency GN-CDE | $\underline{0.777 \pm 0.798}$ | $\underline{1.912 \pm 0.566}$ | $0.802 \pm 0.652$ | $0.909 \pm 0.416$ |
| Pre Mult GN-CDE | $38.717 \pm 6.229$ | $64.825 \pm 14.066$ | $61.226 \pm 47.714$ | $88.503 \pm 74.608$ |
| Original GN-CDE [48] | $\underline{1.060 \pm 0.436}$ | $6.963 \pm 4.698$ | $6.781 \pm 11.159$ | $5.223 \pm 4.240$ |
| PENG-CDE (ours) | $\mathbf{0.525 \pm 0.581}$ | $\mathbf{1.851 \pm 1.488}$ | $\mathbf{0.674 \pm 0.606}$ | $\underline{1.161 \pm 6.011}$ |
| Relative Improvement (%) | $50.52\%$ | $73.41\%$ | $90.07\%$ | $77.77\%$ |

## G  Implementation details

We will release all code alongside the camera-ready version of this paper. The implementation is written in the Python programming language [62], and uses the JAX framework [4]. Key dependencies include Diffrax [34] for differentiable ODE solvers, Equinox [35] for neural network modules in JAX, Optax [4] for optimisation, and Lineax [50] for linear algebra routines. Additional dependencies include NumPy [28], Exca [51], PyTorch [55], and PyTorch Geometric [23].

All differential equation-based models use the Tsitouras' 5/4 method [61] as implemented in Diffrax.

### G.1  Compute resources

All experiments were conducted on a computing cluster equipped with NVIDIA H100 (80 GB) and H200 (141 GB) GPUs. Each node in the cluster features 64-core AMD EPYC 9334 CPUs running at 3.90 GHz, along with 256 GB of RAM. The heat diffusion and gene regulation experiments described

in Section 4.1 required approximately 3.5 GPU days to complete, whereas the oversampling and irregularity experiments finished within a total of 12 GPU hours. The PyTorch Geometric Temporal experiments ran for approximately 8 hours. Running STG-NCDE, GN-CDE, and both versions of our model on the two Temporal Graph Benchmark tasks required around 8 GPU hours in total.

## G.2 Heat diffusion and gene regulation

Two important examples of dynamical systems on networks are heat diffusion and gene regulation dynamics, both of which describe the local exchange of information or state between connected nodes. While heat diffusion models symmetric spreading (e.g., temperature equalisation), gene regulation introduces asymmetric, nonlinear interactions typical of biological systems. These processes can be modelled as follows:

$$\text{Heat Diffusion:} \quad \frac{d\,\mathbf{x}_u(t)}{dt} = \mathcal{L}_t \mathbf{x}_u(t) = \sum_{v \in \mathcal{N}_u(t)} \left( \frac{\mathbf{x}_u(t)}{\sqrt{d_u}} - \frac{\mathbf{x}_v(t)}{\sqrt{d_v}} \right) \tag{24}$$

$$\text{Gene Regulation:} \quad \frac{d\,\mathbf{x}_u(t)}{dt} = -\mathbf{x}_u(t)f + \sum_{v \in \mathcal{N}_u(t)} \frac{\mathbf{x}_v(t)}{\mathbf{x}_v(t) + 1} \tag{25}$$

where $\mathbf{x}_u(t)$ is the temperature of node $u$ at time $t$, $\mathcal{N}_u(t)$ denotes its neighbourhood, and $\mathcal{L}_t$ is the normalised graph Laplacian.

To extend our evaluation to economic networks and social interactions, we additionally include the following two dynamical systems, in which $\mathbf{x}_u(t)$ models the personal capital or opinion of node $u$ at time $t$, respectively:

$$\text{Wealth Dynamics:} \quad \frac{d\,\mathbf{x}_u(t)}{dt} = s_i\,\mathbf{x}_u(t)^{0.6} + \sum_{v \in \mathcal{N}_u(t)} (\mathbf{x}_v(t) - \mathbf{x}_v(t)) + \delta\mathbf{x}_u(t) \tag{26}$$

$$\text{Opinion Dynamics:} \quad \frac{d\,\mathbf{x}_u(t)}{dt} = -\mathbf{x}_u(t) + \text{threshold}\left( \sum_{v \in \mathcal{N}_u(t)} \mathbf{x}_v(t), 0.5 \right). \tag{27}$$

Here, *threshold* denotes the thresholding function defined as $\text{threshold}(x, y) = 1$ iff $x \geq y$ and 0 otherwise.

## G.3 Synthetic experiments

To ensure a fair comparison, the vector fields of all differential equation-based models are implemented using a two-layer GCN [38] with a hidden dimension of 16, and layer normalisation is applied. The models are trained for 2000 epochs using the Adam optimiser [37] with a learning rate of $10^{-2}$ and weight decay of $10^{-4}$.

### G.3.1 SIR model

The Susceptible-Infected-Recovered (SIR) model on graphs [33] is a network-based extension of the foundational compartmental model in epidemiology [32] that describes how infectious diseases spread through a structured population. It partitions nodes in a graph into three groups: susceptible ($S_v$), infected ($I_v$), and recovered ($R_v$) for each node $v$, with transitions governed by the system

$$\frac{dS_v}{dt} = -\beta S_v \sum_{u \in \mathcal{N}(v)} I_u, \quad \frac{dI_v}{dt} = \beta S_v \sum_{u \in \mathcal{N}(v)} I_u - \gamma I_v, \quad \frac{dR_v}{dt} = \gamma I_v, \tag{28}$$

where $\beta$ is the transmission rate, $\gamma$ the recovery rate, and $\mathcal{N}(v)$ denotes the neighbours of node $v$ in the graph.

A key insight from the graph-based SIR model is that the epidemic threshold depends not only on the infection ratio $R = \beta/\gamma$, but also on the structure of the underlying graph, such as its spectral radius or degree distribution. If local connectivity supports sufficient transmission, the infection can spread widely; otherwise, it dies out. Despite its structural complexity, the graph SIR model preserves essential features of epidemic dynamics and extends the classical model to networked populations.

### G.3.2 Oversampling

To study the dependence of performance and compute time on the number of observed samples, we construct grid graphs with $100$ nodes and generate irregularly sampled time series between $T = 0$ and $T = 1$, using $n = 20, 40, 60, \ldots, 200$ observations respectively. The dynamic graph topology is generated as described in Section 4.1. Trajectories are produced by numerically solving Equation 28 with random initial conditions. We use two parameter settings: $(\beta_1 = 0.25, \gamma_1 = 0.7)$, modelling a scenario where the epidemic dies out, and $(\beta_2 = 0.3, \gamma_2 = 0.3)$, modelling a scenario where the infection spreads. For each setting, we generate 50 trajectories each for training, validation, and test sets.

The task is to perform binary classification, predicting whether a given trajectory corresponds to an outbreak (i.e., sustained spread of infection) or a non-outbreak (i.e., rapid die-out).

We train the models using binary cross-entropy loss for 2000 epochs, applying early stopping with a patience of 200 epochs and a minimum of 200 epochs. Training is performed using the Adam optimizer [37] with a learning rate of $10^{-2}$ and a weight decay of $10^{-4}$. The model hyperparameters are summarised in Table 6.

Table 6: Hyperparameters for SIR experiments.

|  | Perm Equiv GN-CDE | DCRNN |
|---|---|---|
| **Hidden Dimension** | 32 | 8 |
| **Number of Layers** | 3 | 3 |
| **Data Embedding Dimension** | 3 | 3 |
| **Layer type** | GCN | – |
| **Chebyshev Order** | – | 3 |

### G.3.3 Irregular observation spacing

To assess how the regularity of observation times affects model performance, we generate classification data exactly as before, except that we now sample the irregular observation time-stamps from a Gamma-driven process. In more detail, let $N$ be the total number of desired time-points on the interval $[0, T]$. We begin by drawing $N - 1$ independent inter-arrival increments

$$\Delta t_i \sim \text{Gamma}(k, \theta), \quad i = 1, \ldots, N - 1,$$

where $k > 0$ controls the shape (and hence burstiness) and $\theta > 0$ scales the raw increments. To force the observations to span exactly $[0, T]$, we normalise the increments by their sum:

$$\tilde{\Delta} t_i = \frac{\Delta t_i}{\sum_{j=1}^{N-1} \Delta t_j} T, \qquad t_i = \sum_{j=1}^{i} \tilde{\Delta} t_j,$$

with $t_0 = 0$ and $t_N = T$. This construction guarantees $0 = t_0 < t_1 < \cdots < t_N = T$ and yields exactly $N + 1$ strictly increasing time-stamps.

In our experiments, we set $T = 1$ and draw $N = 20$ observations per trajectory using this Gamma-based sampler for $k \in \{3, 5, 10, 25, , 50, 100\}$. All other aspects of model training - network architecture, optimisation schedule, and hyperparameters - remain identical to those in the previous section.

### G.4 Real-world experiments

All models are trained over 200 epochs with an early stopping criterion with a patience of 15 and a minimum epoch of 20. In all applications, we utilise cubic splines [34, Chapter 3.5] for interpolation of the data. All hyperparameters have been selected from the ranges in Table 7 using a grid search. For the chosen values see Tables 8, 9 and 10. All experiments have been averaged over 10 random seeds.

Table 7: Ranges the optimal hyperparameters (PENG) GN-CDE (top) and STG-NCDE (bottom).

|  | Range |
|---|---|
| **Hidden Dimension** | $\{8, 16, 32, 64\}$ |
| **Number of Layers** | $\{2, 3, 4\}$ |
| **Data Embedding Dimension** | $\{8, 16, 32, 64\}$ |
| **Number of Recurrent Layers** | $\{1, 2, 3\}$ |
| **Recurrent Hidden Dimension** | $\{8, 16, 32, 64\}$ |
| **Number of GNN Layers** | $\{1, 2, 3\}$ |
| **GNN Hidden Dimension** | $\{8, 16, 32, 64\}$ |
| **Data Embedding Dimension** | $\{8, 16, 32, 64\}$ |
| **Chebychev Order** | $\{2\}$ |
| **Node Embedding Dimension** | $\{5\}$ |

Table 8: Hyperparameters for real-world experiments for the GN-CDE model.

|  | tgbn-trade | tgbn-genre | twitter-tennis | england-covid |
|---|---|---|---|---|
| **Hidden Dimension** | 8 | 8 | 8 | 32 |
| **Number of Layers** | 3 | 2 | 2 | 4 |
| **Data Embedding Dimension** | 16 | 8 | - | - |
| **Layer type** | GCN | GCN | GCN | GCN |

## G.5   Source target identification

In [59], the authors address the well-known issue that machine-learned models underperform on node affinity prediction tasks compared to simple heuristics [53] by modifying the message construction in TGN [53] as follows: Given an interaction between nodes $u$ and $v$ at time $t$ with associated data $e_{uv}(t)$, the messages $\mathbf{m}_u(t)$ and $\mathbf{m}_v(t)$ sent to nodes $u$ and $v$, respectively, are defined as

$$\mathbf{m}_u(t) = \mathbf{msg}_s\big(\mathbf{s}_u(t^-), \mathbf{s}_v(t^-), \phi_t(\Delta t), e_{uv}(t), \phi_n(u), \phi_n(v)\big), \tag{29}$$

$$\mathbf{m}_v(t) = \mathbf{msg}_d\big(\mathbf{s}_v(t^-), \mathbf{s}_u(t^-), \phi_t(\Delta t), e_{uv}(t), \phi_n(v), \phi_n(u)\big), \tag{30}$$

where $\Delta t$ is the time elapsed since their last interaction and $\phi : \mathbb{R} \to \mathbb{R}^n$ is an encoder function for node indices. In their work, the authors use $\phi(u) = (\cos(\omega_i u))_{i=0}^{n-1}$.

Inspired by this, we incorporate source-target identification into our equivariant fusion by transforming the adjacency matrix $\mathbf{A}_s$ and its derivative $\frac{d\mathbf{A}_s}{ds}$ using an attention-style approach. Specifically, each entry $a_{uv}$ of either matrix is passed through an MLP together with the encoding of both $u$ and $v$ to form a new matrix $\mathbf{B}$ with entries

$$b_{uv} = \mathtt{MLP}(a_{uv}||\phi(u)||\phi(v)). \tag{31}$$

where $||$ denotes concatenation. We selected the following hyperparameters: an embedding dimension of $n = 512$, an MLP width of 8, and 2 hidden layers. We slightly deviated from the original approach by defining $\phi(u) = \mathtt{MLP}(u)$ with a hidden dimension of 8 and 2 hidden layers.

Table 9: Hyperparameters for real-world experiments for the Permutation Equivariant GN-CDE model.

|  | tgbn-trade | tgbn-genre | twitter-tennis | england-covid |
|---|---|---|---|---|
| **Hidden Dimension** | 32 | 16 | 64 | 64 |
| **Number of Layers** | 2 | 3 | 3 | 3 |
| **Data Embedding Dimension** | 8 | 8 | - | - |
| **Layer type** | GCN | GCN | GCN | GCN |

Table 10: Hyperparameters for real-world experiments for the STG-NCDE model.

|  | tgbn-trade | tgbn-genre | twitter-tennis | england-covid |
|---|---|---|---|---|
| **Number of Recurrent Layers** | 1 | 3 | 3 | 1 |
| **Recurrent Hidden Dimension** | 8 | 16 | 32 | 32 |
| **Number of GNN Layers** | 2 | 1 | 2 | 3 |
| **GNN Hidden Dimension** | 8 | 16 | 32 | 32 |
| **Data Embedding Dimension** | 32 | 8 | - | - |
| **Chebychev Order** | 2 | 2 | 2 | 2 |
| **Node Embedding Dimension** | 5 | 5 | 5 | 5 |

Table 11: Runtime per training epoch ($s$) for GN-CDE and PENG-CDE across different node counts in Section 4.1.

| Model | 128 | 256 | 512 | 1024 | 2048 |
|---|---|---|---|---|---|
| Pre Mult GN-CDE [48] | 0.45 | 0.56 | 1.18 | 1.62 | 9.15 |
| PENG-CDE (ours) | 0.42 | 0.49 | 0.57 | 0.90 | 1.50 |
| Rel. Improv. (%) | 7.1 | 14.3 | 107.0 | 80.0 | 510.0 |

## H   Statistical analysis of Experiments

In this section, we conduct a statistical verification of the relevance of our experiments. Note that in Section 4 whenever possible, we reported means along with $95\%$ confidence intervals. This applies to all synthetic experiments and the PyTorch Geometric Temporal datasets reported in Tables 1 and 2 in the main text. For the Temporal Graph Benchmark datasets, baseline results were only available as mean NDGC@10 scores with standard deviations, so we adopted the same format for consistency.

To address statistical significance testing, we employ critical difference diagrams, based on a two-stage procedure:

1. A global Friedman test was used to determine whether any model differences were statistically significant.

2. If the null hypothesis was rejected, we performed pairwise comparisons using the aligned Friedman post-hoc test.

We present the critical difference diagrams for the dynamical systems, Pytorch Geometric Temporal and Temporal Graph Benchmark datasets in Figures 3, 4 and 5, respectively.

We can see that in all synthetic experiments (Figure 3), our PENG-CDE consistently ranks within the top non-significant clique, which demonstrates its strong and stable performance. For the PyTorch Geometric Temporal datasets (twitter-tennis, england-covid; Figure 4), the global Friedman test was inconclusive, meaning the test did not reject the null hypothesis that no model (even the baselines) is statistically different from the others. On the Temporal Graph Benchmark, the critical difference analysis (Figure 5) shows that PENG-CDE performs significantly better than both GN-CDE and STG-NCDE.

## I   Runtime analysis

**Runtime.** To highlight the computational benefits of our approach, we repeat the experiments from Section 4.1 with varying numbers of nodes and record the time per training epoch in Table 11. Our approach exhibits significantly better runtime performance scaling, as the number of learnable parameters in the fusion matrices $\mathbf{W}_i$ in the Pre Mult GN-CDE scales quadratically with the number of nodes, whereas the parameter count of PENG-CDE is independent of graph size.

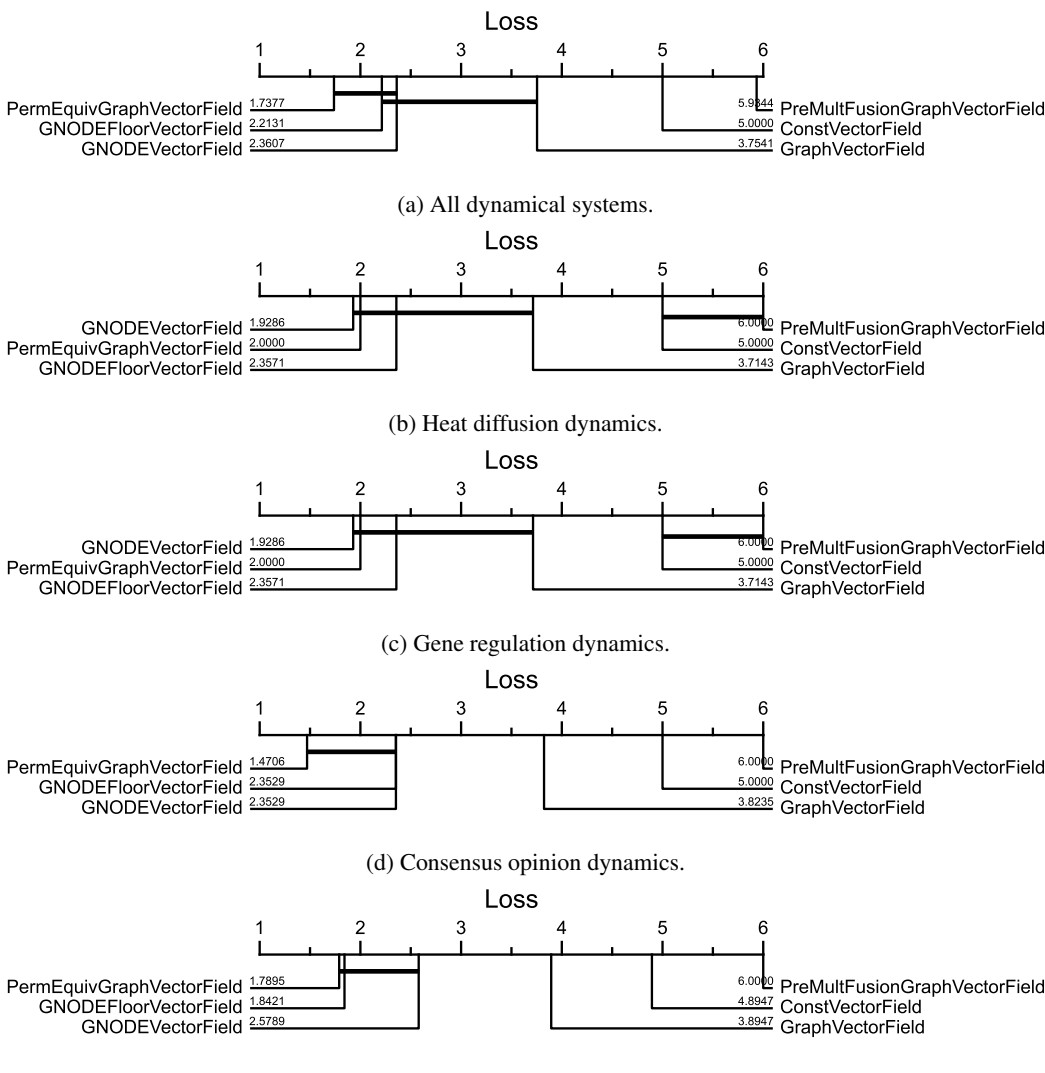

(a) All dynamical systems.

(b) Heat diffusion dynamics.

(c) Gene regulation dynamics.

(d) Consensus opinion dynamics.

(e) Personal capital dynamics.

Figure 3: Critical difference diagrams for the synthetic experiments in Section 4.1 and Tables 1 and 5.

## J  Miscellaneous

### J.1  Societal impact

This work constitutes foundational research and is not tied to any specific application or deployment. As such, it shares the general risks and benefits inherent to novel machine-learning architectures. For example, dynamic graph representation learning models have traditionally been applied to tasks such as traffic forecasting [11, 10] and molecular modelling [58], and we hope that the advances presented here will further drive progress in these and related areas. Overall, we assess the societal impact of this work to be predominantly positive.

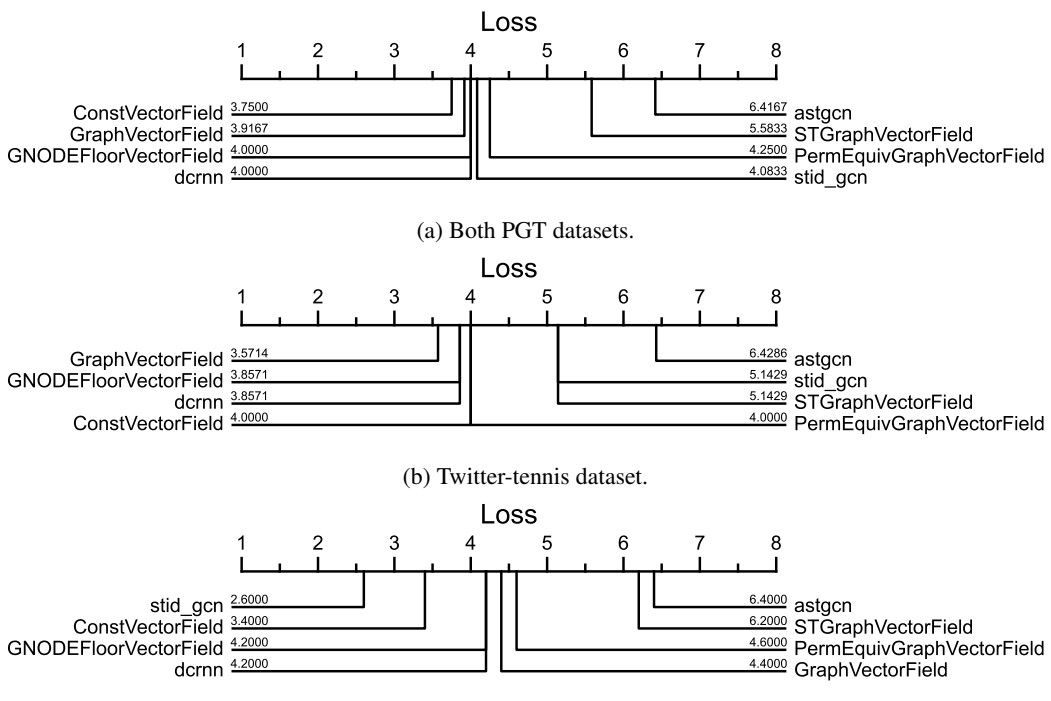

(a) Both PGT datasets.

(b) Twitter-tennis dataset.

(c) England-covid dataset.

Figure 4: Critical difference diagrams for the Pytorch Geometric Temporal (PGT) real-world experiments in Section 4.2 and Table 2.

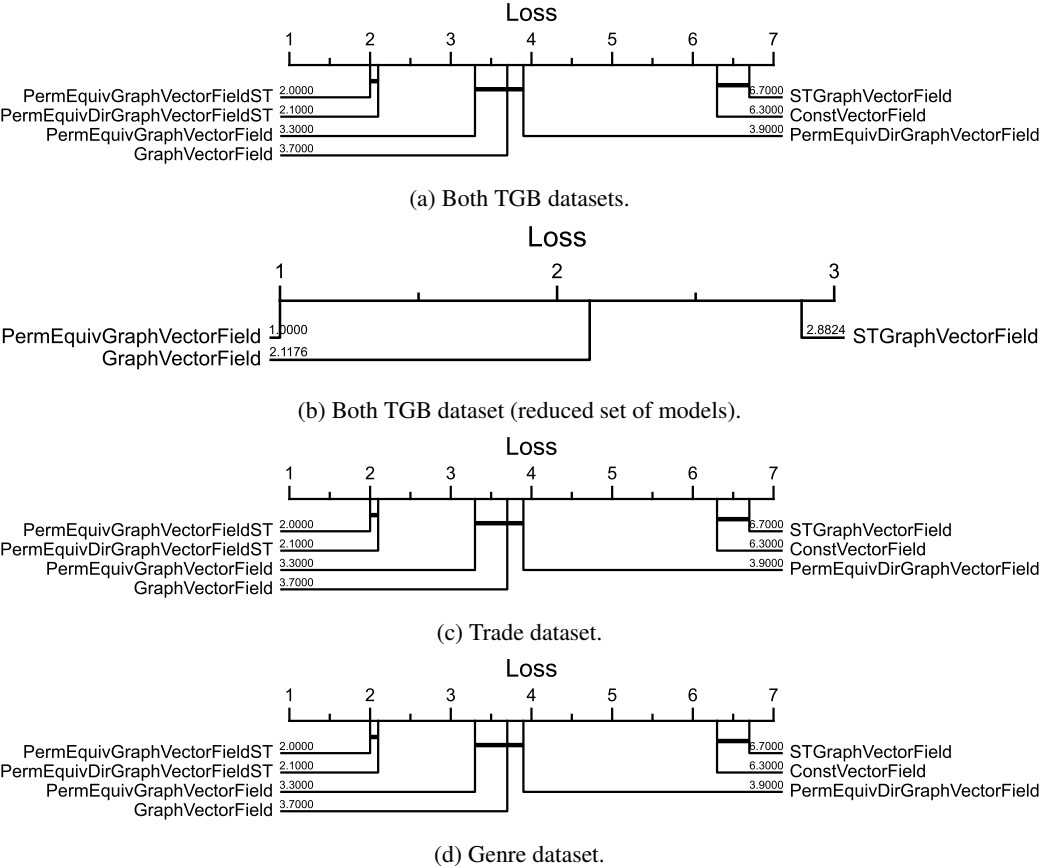

(a) Both TGB datasets.

(b) Both TGB dataset (reduced set of models).

(c) Trade dataset.

(d) Genre dataset.

Figure 5: Critical difference diagrams for the Temporal Graph Benchmark (TGB) real-world experiments in Section 4.2 and Table 3.

