# OpenReview forum: "Permutation Equivariant Neural Controlled Differential Equations for Dynamic Graph Representation Learning"
_NeurIPS.cc/2025/Conference — NeurIPS 2025 poster_

### Official Review · Reviewer_FW39 · 2025-06-16

**Clarity:** 3
**Significance:** 2
**Originality:** 3
**Rating:** 4
**Confidence:** 4

**Summary:**

The paper introduces Permutation Equivariant Neural Graph Controlled Differential Equations (PENG-CDEs), extending Graph Neural CDEs to enforce permutation equivariance, reducing parameter count while preserving representation power. It proves optimality within permutation-equivariant function spaces (Theorem 3.1) and equivariance under time reparametrizations and node permutations (Proposition 3.2). Evaluated on synthetic (heat diffusion, gene regulation, SIR model) and real-world tasks (TGB-genre, tgbn-trade, twitter-tennis, england-covid), PENG-CDEs outperform baselines like DCRNN, STG-NCDE, and GN-CDE, setting a state-of-the-art on TGB-genre node affinity prediction.

**Questions:**

1. Metrics are averaged over 10 seeds but lack statistical tests. Can you include t-tests or confidence intervals to confirm PENG-CDEs’ superiority, especially for small margins (e.g., tgbn-trade)?
Suggestion: Add statistical tests in Section 4. Significant improvements could raise the Quality.

2. Evaluation is limited to moderate-sized graphs. Can you test PENG-CDEs on large-scale dynamic graphs (e.g., million-node networks) to demonstrate scalability?
Suggestion: Include scalability experiments in Section 4.4. Demonstrated scalability could raise the Significance.

3. Synthetic tasks focus on SIR and diffusion models. Can you evaluate PENG-CDEs on diverse graph dynamics (e.g., economic networks, social interactions)?
Suggestion: Add diverse synthetic tasks in Section 4.1. Broader applicability could raise the Significance.

4. Theorem 3.1 assumes simplified conditions (e.g., linear maps). How robust is PENG-CDEs to complex, non-linear graph dynamics? Can you provide analysis?
Suggestion: Analyze robustness in Appendix D or Section 3.2.

5. Hyperparameters and fusion details are in Appendix F. Can you include key implementation details in Section 3 to improve readability?
Suggestion: Add a paragraph in Section 3 summarizing the architecture and training.

**Ethical Concerns:**

["NO or VERY MINOR ethics concerns only"]

**Final Justification:**

There’s still room for improvement in these areas, but your response has addressed some of the issues, so I’ll give a score.

**Limitations:**

Yes

**Paper Formatting Concerns:**

Minor issues, including inconsistent table captions, font sizes, and a potential anonymity concern, should also be addressed to enhance compliance and readability.

**Quality:**

3

**Strengths And Weaknesses:**

Strengths:
Theorem 3.1 (Section 3.2) proves PENG-CDEs’ optimality in permutation-equivariant spaces, and Proposition 3.2 ensures equivariance, with complete proofs in Appendices D–E. Assumptions are clear.
Evaluated on synthetic tasks (heat diffusion, gene regulation, SIR model, Section 4.1) and four real-world datasets (Section 4.4), PENG-CDEs outperform baselines (e.g., DCRNN, STG-NCDE, Table 9). Ablations (Appendix H) analyze fusion weights.
While PENG-CDEs set a state-of-the-art on TGB-genre, gains over GN-CDE are small in some tasks (e.g., tgbn-trade, twitter-tennis, Section 4). No statistical significance tests (e.g., t-tests) confirm superiority, despite averaging over 10 seeds.
Theorem 3.1 assumes simplified conditions (e.g., linear equivariant maps, finite group, Appendix D), potentially unrealistic for complex dynamic graphs. The proof relies on standard projection theory (Appendix B), offering limited novelty.
Real-world datasets are limited to four, with no large-scale graphs (e.g., million-node networks). Scalability is untested despite dynamic graph complexity. Synthetic tasks (SIR, heat diffusion) are controlled, lacking diversity in graph dynamics (e.g., non-epidemic models).
Baselines like DCRNN and STG-NCDE use grid-searched hyperparameters (Table 5), but modern temporal GNNs (e.g., GraphMixer [14]) are not fully compared, potentially inflating PENG-CDEs’ performance.

Clarity:
The paper is organized with clear sections on background, methodology, theory, and experiments. Appendices A–H provide detailed proofs and implementation.
Hyperparameters (e.g., GCN layers, learning rate) and fusion operation specifics are in Appendix F, reducing standalone clarity. The MLP in source-target identification (Appendix F.5) lacks architectural detail.
Terms like “permutation equivariant function spaces” and “Haar measure” (Appendix C) are mathematically dense, lacking intuitive explanations for non-experts.
Section 4 reports metrics but lacks analysis of why PENG-CDEs excel on TGB-genre but not others. Ablation insights (Table 9) are descriptive, not mechanistic.

Significance:
Dynamic graph learning is critical for traffic forecasting, molecular modeling, and epidemiology (Appendix G.1). PENG-CDEs’ efficiency could benefit these fields.
Gains over GN-CDE are modest, and broader applicability (e.g., large-scale graphs) is untested. The practical advantage is limited without statistical validation.
The focus on node/graph-level tasks excludes other dynamic graph problems (e.g., link prediction, clustering).
Appendix F.1 notes moderate compute (8 GPU hours for TGB tasks), but scalability to massive graphs is unaddressed, critical for real-world applications.
Appendix G.1 claims positive impact but lacks specific negative impact analysis (e.g., misuse in surveillance, Checklist Q10), weakening assessment.

Originality:
PENG-CDEs extend GN-CDEs with permutation equivariance, a new approach reducing parameters while preserving power (Section 3).
However, PENG-CDEs build on GN-CDEs [10] and Neural ODEs [7], with equivariance concepts from Maron et al. [44]. The contribution is an adaptation, not a groundbreaking paradigm. Theoretical results rely on standard tools (e.g., projection theorem, Haar measure, Appendix B–C), limiting novelty.

---

> ### Author Rebuttal · Authors · 2025-07-31
>
> We sincerely thank the reviewer for their detailed and constructive feedback. We begin by responding to the specific questions and will then address your remaining concerns.
>
> **[Q1]: Statistical significance tests**
>
> Whenever possible, we reported means along with 95% confidence intervals. This applies to all synthetic experiments and the PyTorch Geometric Temporal datasets reported in Tables 1 and 2 of the main paper. For the Temporal Graph Benchmark datasets, baseline results were only available as mean NDGC scores with standard deviations, so we adopted the same format for consistency.
>
> To address statistical significance testing, we employ critical difference diagrams, based on a two-stage procedure:
>
> 1.  A global Friedman test  was used to determine whether any model differences were statistically significant.
>
> 2.  If the null hypothesis was rejected, we performed pairwise comparisons using the aligned Friedman post-hoc test.
>
> We present the results in Tables 1-6 below and will include the full CD diagrams in the camera-ready version of the manuscript.
>
> Table 1: Critical difference diagram for heat diffusion synthetic experiment.
> |Mean Rank|Model|Non‑significant clique(s)|
> |----------:|------------------|--------------------------:|
> |1.93|Graph Neural ODE|A|
> |2.00|PENG‑CDE|A|
> |2.36|Adjacency GN‑CDE|A|
> |3.71|GN‑CDE|A|
> |5.00|Constant|B|
> |6.00|Pre Mult GN‑CDE|B|
>
> Table 2: Critical difference diagram for gene regulation synthetic experiment.
> |Mean Rank|Model|Non‑significant clique(s)|
> |----------|------------------|--------------------------|
> |1.73|PENG‑CDE|A|
> |2.45|Adjacency GN‑CDE|A|
> |2.55|Graph Neural ODE|A|
> |3.45|GN‑CDE|A|
> |5.18|Constant|B|
> |5.64|Pre Mult GN‑CDE|C|
>
> Table 3: Critical difference diagram for opinion dynamics synthetic experiment.
> |Mean Rank|Model|Non‑significant clique(s)|
> |----------|------------------|--------------------------|
> |1.47|PENG‑CDE|A|
> |2.35|Adjacency GN‑CDE|A|
> |2.35|Graph Neural ODE|A|
> |3.82|GN‑CDE|B|
> |5.00|Constant|C|
> |6.00|Pre Mult GN‑CDE|D|
>
> Table 4: Critical difference diagram for wealth formation synthetic experiment.
> |Mean Rank|Model|Non‑significant clique(s)|
> |----------|------------------|--------------------------|
> |1.79|PENG‑CDE|A|
> |1.84|Adjacency GN‑CDE|A|
> |2.58|Graph Neural ODE|A|
> |3.89|GN‑CDE|B|
> |4.89|Constant|C|
> |6.00|Pre Mult GN‑CDE|D|
>
> Table 5: Critical difference diagram for PGT experiments.
> |Mean Rank|Model|Non‑significant clique(s)|
> |----------|----------------------|---------------------|
> |3.75|Constant|A|
> |3.92|GN-CDE|A|
> |4.00|Adjacency GN-CDE|A|
> |4.00|DCRNN|A|
> |4.08|STIDGCN|A|
> |4.25|PENG-CDE|A|
> |5.58|STG-NCDE|A|
> |6.42|ASTGCN|A|
>
> Table 6: Critical difference diagram for TGB experiments.
> |Mean Rank|Model|Non-significant clique|
> |----------|----------------|---------------------|
> |1.00|PENG-CDE|A|
> |2.15|GN-CDE|B|
> |2.85|STG-NCDE|C|
>
> We can see that in all synthetic experiments (Tables 1-4), our PENG-CDE consistently ranks within the top non-significant clique which demonstrates its strong and stable performance. For the PyTorch Geometric Temporal datasets (twitter-tennis, england-covid; Table 5), the global Friedman test was inconclusive, meaning the test did not reject the null hypothesis that no model (even the baselines) is statistically different from the others. On the Temporal Graph Benchmark, the critical difference analysis (see Table 6) shows that PENG-CDE performs significantly better than both GN-CDE and STG-NCDE.
>
> **[Q2]: Evaluation on larger graphs**
>
> The main scope of our paper is the derivation of a new mathematical formalism for dynamic graph representation learning. Our method belongs to a family of models that are unique in their approach to tackling graph dynamics. Most other approaches are permutation equivariant due to handling the graph structure using an existing graph neural network structure.
>
> Our implementation relies on JAX and, in particular, the Diffrax library. As both currently lack full support for sparse matrix operations, PENG-CDE, like the original GN-CDE, exhibits quadratic memory complexity in the number of nodes. This constrains scalability beyond moderately sized graphs, as discussed in the Limitations section. We view this as an exciting direction for future work. As sparse support in JAX matures beyond its experimental stage, we expect to overcome this scalability bottleneck. Importantly, as shown in Table 1 in our response to Reviewer GJRF, once memory constraints are addressed, PENG-CDE exhibits significantly better runtime scaling than GN-CDE, highlighting its potential for efficient execution even at larger scales.
>
> **[Q3]: More diverse synthetic experiments**
>
> We are happy to provide the results of testing our PENG-CDE, the baselines included in the original submission, as well as the  additional baselines suggested by Reviewer _Dvhd_, in the two tables below.
>
> As suggested by the reviewer, we have added two more dynamical systems to our synthetic experiments: an economic network model and a social interaction network model.
>
> 1.  Capital Gains Wealth Dynamics
>
> 	$\frac{dx_i}{dt} = \delta x_i + s_i x_i^k + \sum_{j} A_{ij} (x_j - x_i)$
>
>
> 2. Nonlinear Opinion Dynamics
>
> 	$\frac{dx_i}{dt} = -x_i + \sigma\left( \sum_{j} A_{ij} \cdot x_j \right)$
>     where  $\sigma$  is a thresholding function.
>
> For the results see Tables 7 and 8. We can see that PENG-CDE outperforms the original GN-CDE by 35–90%, and achieves the overall lowest mean MSE in 6 out of 8 dataset configurations.
>
> **Table 7: Comparison of GN-CDE variants and extended baselines on the wealth dynammics tasks. Mean MSEs with $95\%$ confidence intervals are reported, with the best mean highlighted in bold.**
>
> |Model|Community|Grid|Power Law|Small World|
> |----------------------------|-----------------|--------------------|--------------------|--------------------|
> |DCRNN|2.639±1.981|22.619±17.115|113.564±236.126|200.486±280.387|
> |STIDGCN|1.862±0.947|7.296±2.547|2.124±1.012|3.133±1.286|
> |ASTGCN|3.431±1.224|19.694±2.986|5.153±1.184|9.246±2.204|
> |STG-NCDE|36.087±2.454|44.677±3.755|37.148±2.742|41.046±2.205|
> |Constant|2.504±0.473|14.157±1.746|3.682±0.873|7.445±1.200|
> |Graph Neural ODE (GNODEs)|0.878±0.388|**1.898±1.342**|0.950±0.660|0.940±0.185|
> |Adjacency GN-CDE|0.904±0.493|2.193±1.721|1.075±0.847|1.297±0.765|
> |Pre Mult GN-CDE|7.740±1.122|24.522±6.594|39.715±32.698|16.568±6.902|
> |Original GN-CDE|1.577±0.781|6.582±3.273|2.406±0.965|3.076±1.193|
> |**PENG-CDE (ours)**|**0.522±0.316**|4.273±3.818|**0.863±0.330**|**0.813±0.627**|
> |**Relative Improvement (%)**|66.89%|35.09%|64.13%|73.57%|
>
>
> **Table 8: Comparison of GN-CDE variants and extended baselines on the opinion dynammics tasks. Mean MSEs with $95\%$ confidence intervals are reported, with the best mean highlighted in bold**.
> |Model|Community|Grid|Power Law|Small World|
> |----------------------------|--------------------|--------------------|--------------------|--------------------|
> |DCRNN|20.659±32.149|92.165±132.026|91.687±81.638|173.554±183.913|
> |STIDGCN|2.434±1.562|6.977±2.451|2.724±1.530|3.933±1.567|
> |ASTGCN|22.001±1.519|34.855±2.370|22.517±1.490|27.439±1.554|
> |STG-NCDE|57.495±9.245|85.405±6.479|59.446±9.294|68.092±8.988|
> |Constant|22.989±1.372|33.124±1.275|23.775±1.480|27.239±1.929|
> |Graph Neural ODE (GNODEs)|0.665±0.620|2.338±1.277|3.073±6.114|**0.779±0.417**|
> |Adjacency GN-CDE|0.777±0.798|1.912±0.566|0.802±0.652|0.909±0.416|
> |Pre Mult GN-CDE|38.717±6.229|64.825±14.066|61.226±47.714|88.503±74.608|
> |Original GN-CDE (Qin et al.)|1.060±0.436|6.963±4.698|6.781±11.159|5.223±4.240|
> |**PENG-CDE (ours)**|**0.525±0.581**|**1.851±1.488**|**0.674±0.606**|1.161±6.011|
> |**Relative Improvement (%)**|50.52%|73.41%|90.07%|77.77%|
>
>
> **[Q4]: Non-linear graph dynamics in Theorem 3.1**
>
> The reviewer is correct to point out that Theorem 3.1  is stated under the assumption of a  _linear_  vector field, i.e. the absence of non-linearities. The purpose of this result is to theoretically motivate the architectural modifications we introduced when transitioning from GN-CDEs to PENG-CDEs. Specifically, it formalises the idea that  _“PENG-CDE is the closest equivariant version of GN-CDE we can construct in the linear setting.”_
>
> In practice, however, we train all CDE-based models with non-linearities in the vector field. This enhances their expressive power and enables them to learn the non-linear dynamical systems in Section 4.1 and Tables 7 and 8 above.
>
> **[Q5]: More information on architectures and hyperparameters in main text**
>
> We agree with the reviewer the paper would benefit from more details in the main manuscript and will include this with the camera-ready version.
>
> **[W1]: Lack of novelty**
>
> See our reply to [W1] of reviewer _G4dS_.
>
> **[W2]: Hyperparams of baselines**
>
> To ensure a fair comparison, we followed the hyperparameter settings from the original papers as closely as possible for all baselines, in particular DCRNN and GraphMixer.
>
> **[W3]: Ablation in Table 9 is descriptive, not mechanistic**
>
> We thank the reviewer for pointing this out. We will include additional explanations in the final version. For example, summing over rows or columns of the adjacency matrix corresponds to computing node degrees, indicating that our model can utilise global information such as degree normalisation.
>
> **[W4]: No evaluation of link prediction or clustering**
>
> To ensure a fair comparison with the main baseline GN-CDE, we focused on the setup evaluated in the original paper, which included both graph-level and node-level tasks. This is a promising avenue for future work.
>
> ---
>
> [1] Weilin Cong, Si Zhang, Jian Kang, Baichuan Yuan, Hao Wu, Xin Zhou, Hanghang Tong, and
> Mehrdad Mahdavi. Do we really need complicated model architectures for temporal networks?,
> 2023.
>
> [2] Shenyang Huang, Farimah Poursafaei, Jacob Danovitch, Matthias Fey, Weihua Hu, Emanuele Rossi, Jure Leskovec, Michael Bronstein, Guillaume Rabuseau, and Reihaneh Rabbany. Temporal graph benchmark for machine learning on temporal graphs, 2023

---

> > ### Comment · Reviewer_FW39 · 2025-08-06
> >
> > Several key issues remain in my assessment:
> >
> > While Theorem 3.1 motivates the architecture, its linear assumptions (as acknowledged) limit applicability to non-linear real-world dynamics. The rebuttal clarifies this but doesn't extend the theory (e.g., to non-linear cases), keeping the contribution an adaptation of GN-CDEs [10] and Neural ODEs [7] without groundbreaking innovations. Equivariance draws from Maron et al. [44], reducing novelty.
> >
> > The new synthetic tasks (wealth/opinion dynamics) add diversity, and PENG-CDEs excel, but the focus remains on moderate graphs. The rebuttal emphasizes the paper's mathematical formalism but doesn't address scalability empirically (e.g., via approximations or downsampled large graphs), despite JAX/Diffrax limitations. This leaves practical deployment in large-scale applications (e.g., million-node networks) unproven, a core concern for dynamic graph learning.
> >
> > CD diagrams are valuable and show PENG-CDEs in top cliques, but for PGT datasets (Table 5), the inconclusive Friedman test suggests no significant differences even from baselines, tempering superiority claims. Modern baselines (e.g., GraphMixer [1]) are now referenced but not re-evaluated with the new stats, and hyperparameter alignment is noted but could inflate gains if not fully optimized.
> >
> > Explanations for ablations (e.g., row/column sums capturing degrees) are helpful, but mechanistic analysis (e.g., why PENG-CDEs handle irregularity better) is still descriptive. Adding these to the main text would improve, but the rebuttal doesn't resolve all gaps (e.g., MLP details in Appendix F.5).

---

> > > ### Author Response · Authors · 2025-08-06
> > >
> > > [I1] We would like to clarify the role of the linearity assumption in Theorem 3.1, as it may have been misunderstood. We use this result solely to motivate replacing the unrestricted linear fusion with its permutation-equivariant projection; when those are the only terms under consideration, this is the unique optimal choice. Proposition 3.2 then shows that inserting the same projected fusion back into the full non-linear GN-CDE stack preserves both permutation and time-warp equivariance. Hence the linear assumption is strictly local to the fusion layer and does not restrict the applicability of PENG-CDEs to non-linear dynamics, as confirmed empirically in Section 4. We will revise the manuscript to clarify this point.
> > >
> > > [I2] We appreciate the reviewer's recognition of our contribution in deriving a new mathematical formalism for dynamic graph representation learning. In our response to reviewer GJRF01 (see Table 1), we have provided scalability experiments on modestly sized graphs. Our results show that, where graphs can be accommodated in memory (a current limitation due to JAX/Diffrax), PENG-CDEs offer significantly improved runtime efficiency compared to GN-CDEs. We acknowledge that this currently limits applicability to million-node networks, but we are confident that - using the same formalism - future work can address these scalability constraints.
> > >
> > > [I3] For the PGT datasets, the application of the global Friedman test indicates that no statistically significant differences were observed between any of the models evaluated, including both our proposed methods and all baselines. We interpret this as a limitation of the datasets and the specific evaluation setup, rather than a reflection of any particular model's strengths or weaknesses. Consequently, we conclude that this suite of datasets is not sufficient to statistically establish the superiority of any approach under the current experimental conditions. Nevertheless, if one considers mean performance alone (i.e., without regard to statistical significance), PENG-CDE achieves the best average results in test loss. As this limitation only became apparent during the rebuttal phase, we will revise the claims in the camera-ready version of the paper accordingly.
> > > We would also like to emphasise that on all other tested datasets, in particular the TGB dataset suite, PENG-CDE is statistically significantly better than the baselines.
> > >
> > > [I4] Could you please clarify what you mean by "modern baselines are now referenced but not re-evaluated with the new stats"? We specifically compare against GraphMixer in Section 4.2. To address potential concerns about fair comparison and hyperparameter tuning, we outline our approach as follows:
> > > - Synthetic tasks: All models were trained using identical hyperparameters (e.g., number of hidden layers, hidden dimension).
> > > - PGT real-world experiments: We performed hyperparameter sweeps over the same grid for all models compared, ensuring a fair evaluation.
> > > - TGB real-world experiments: For all baselines, we used the reported values from the original TGB paper [1], the official leaderboard, or [2]. In both papers, a comprehensive hyperparameter sweep was conducted for each model; for example, [2] specifically notes that for GraphMixer, they "perform an exhaustive grid search to find the optimal configurations of some critical hyperparameters for more reliable comparisons". The only non-GNCDE model not previously implemented for TGB datasets is STG-NCDE; for this model, we performed a hyperparameter sweep over the same region as for PENG-CDE and GN-CDE.
> > > We acknowledge that these details could be highlighted more clearly in the main text and will add further clarification in the camera-ready version.

---

> ### Author Response · Authors · 2025-08-06
>
> [I5] We are happy to expand on subsection "FINDING IV: PENG-CDE is robust to oversampling and irregular sampling" in Section 4.1, providing a mechanistic interpretation for the robustness of PENG-CDEs in these scenarios:
> "CDE-based models such as PENG-CDE decouple the computational complexity of their forward passes from the number of input observations: the number of vector field evaluations is determined by the ODE solver, not by the sampling rate of the data. This property makes them inherently robust to oversampling. As shown in Figure 2, increasing the sampling frequency negatively impacts both the performance and runtime of the recurrent baseline (DCRNN), while our model remains largely unaffected. Although STIDGCN also maintains performance under oversampling, its computational cost increases with the number of observations, in contrast to the constant runtime of our approach. Likewise, PENG-CDE sustains high performance across all levels of sampling irregularity, while the other models exhibit degraded performance."
>
> Regarding the MLP, we will add the following to Section F.5 after Equation 29:
> "We selected the following hyperparameters: an embedding dimension of $n = 512$; an Euclidean MLP with input dimension $1 + n + n$, width $8$, two hidden layers, ReLU activation functions, and output dimension $1$."
>
> ---
>
> [1] Shenyang Huang, Farimah Poursafaei, Jacob Danovitch, Matthias Fey, Weihua Hu, Emanuele Rossi, Jure Leskovec, Michael Bronstein, Guillaume Rabusseau, and Reihaneh Rabbany. Temporal graph benchmark for machine learning on temporal graphs. Advances in Neural Information Processing Systems, 2023
>
> [2] Le Yu, Leilei Sun, Bowen Du, Weifeng Lv. Towards Better Dynamic Graph Learning: New Architecture and Unified Library. Advances in Neural Information Processing Systems, 2023

---

### Official Review · Reviewer_Dvhd · 2025-07-01

**Clarity:** 3
**Significance:** 2
**Originality:** 2
**Rating:** 4
**Confidence:** 4

**Summary:**

This paper proposes Permutation Equivariant Neural Graph Controlled Differential Equations (PENG-CDE), an advancement over existing Neural Graph CDE methods. PENG-CDE ensures permutation equivariance to enhance generalization capabilities. Experiments conducted on synthetic and real-world datasets demonstrate PENG-CDE's performance advantages over other dynamic graph representation methods.

**Questions:**

The Questions are listed in Weaknesses.

**Ethical Concerns:**

["NO or VERY MINOR ethics concerns only"]

**Final Justification:**

The feedback fulfils my concerns. I will increase my score.

**Limitations:**

Yes

**Paper Formatting Concerns:**

No paper formatting concerns.

**Quality:**

2

**Strengths And Weaknesses:**

Strengths:

1. **Novel Approach:** The idea of integrating a permutation equivariant map into Neural Graph CDE for handling edge-valued signals is innovative.

2. **Robust Theoretical Foundation:** The paper includes rigorous and detailed theoretical proofs confirming the permutation equivariance of the proposed model.

3. **Clear Presentation:** The manuscript is well-organized, clearly written, and easy to follow.

Weaknesses:

1. **Limited Real-world Performance:** According to Tables 1, 2, and 3, while PENG-CDE performs strongly in synthetic tasks, it demonstrates only average performance in real-world scenarios (e.g., england-covid, twitter-tennis, and tgbn-trade). The marginal superiority of PENG-CDE over other methods on these tasks raises concerns about the practical effectiveness of the proposed approach.

2. **Missing Relevant Baselines:** The absence of critical baselines such as STG-NCDE [1], DCRNN [2], and STIDGCN [3] in the synthetic experiments (heat diffusion and gene regulation) weakens the validity and comprehensiveness of these results.

3. **Minor Grammatical Errors:** A grammatical mistake was found at line 173: "Projecting an the flow of an ODE" should be corrected to "Projecting the flow of an ODE."

---

> ### Author Rebuttal · Authors · 2025-07-31
>
> We appreciate the reviewer’s concerns and thank them for their thoughtful feedback. We will address the points raised one by one.
>
> **[W1] Limited real-world performance**
>
> First, we would like to highlight that across all tested real-world datasets - twitter-tennis, england-covid, tgbn-trade, and tgbn-genre - PENG-CDE achieves the best mean performance  among all baselines in terms of mean losses and NDCG@10, respectively. On tgbn-genre, a widely used benchmark, our method sets a new state-of-the-art result.
>
> Regarding the reviewer's concern about marginal performance differences: in our response to  [Q1] of Reviewer _FW39_, we perform a statistical significance analysis using Friedman tests followed by  critical difference diagrams. We kindly refer the reviewer to Tables 1–6 in our reply, which show that while none of the models (including the baselines) on twitter-tennis and england-covid are statistically distinguishable by this test, PENG-CDE's superior performance over both GN-CDE and STG-NCDE on the Temporal Graph Benchmark (TGB) and our dynamical systems datasets are statistically significant.
>
> Secondly, beyond strong empirical results, PENG-CDE offers substantial practical advantages over the original GN-CDE:
> -  PENG-CDE uses only ~2% more parameters than GN-CDE, yet reduces the average MSE by 0.7% and 5.4% on twitter-tennis and england-covid datasets, respectively, and improves NDCG@10 by 2.5% and 13.7% on the TGB datasets.
> -   Compared to the Pre-Mult GN-CDE variant, PENG-CDE reduces its parameter count by over 99%, while still reducing the mean MSE by a factor of 8.56× to 27.65× on the synthetic datasets.
>
> **[W2] Missing baselines in synthetic experiments**
>
> We thank the reviewer for suggesting the inclusion of additional baselines in the synthetic experiments. Please refer to Tables 1 and 2 below for updated results incorporating the requested baselines. As shown, our method consistently outperforms these baselines, often by an order of magnitude. For further synthetic experiments on wealth formation and opinion dynamics including these baselines, we kindly refer to our response to Reviewer _FW39_.
>
> **Table 1: Comparison of GN-CDE variants and extended baselines on the heat diffusion tasks. Mean MSEs with 95% confidence intervals are reported, with the best mean highlighted in bold.**
> | Model                        | Community         | Grid              | Power Law            | Small World          |
> | ---------------------------- | ----------------- | ----------------- | -------------------- | -------------------- |
> | **DCRNN**                        | 0.722 ± 1.145     | 70.392 ± 108.473  | 1.690 ± 1.456        | 84.690 ± 105.696     |
> | **STIDGCN**                      | 0.554 ± 0.393     | 4.297 ± 0.975     | 0.907 ± 0.268        | 1.826 ± 0.240        |
> | **ASTGCN**                       | 2.188 ± 0.656     | 15.480 ± 1.757    | 3.849 ± 0.832        | 8.269 ± 1.094        |
> | **STG-NCDE**     | 2.091 ± 0.645     | 11.989 ± 1.090    | 3.518 ± 1.105        | 6.902 ± 1.303        |
> | Constant                     | 1.936 ± 0.550     | 11.155 ± 0.669    | 3.147 ± 0.850        | 6.286 ± 1.056        |
> | Graph Neural ODE (GNODEs)    | 0.237 ± 0.322     | 1.001 ± 0.751     | 0.270 ± 0.310 | 0.311 ± 0.268 |
> | Adjacency GN‑CDE             | 0.208 ± 0.240     | 0.691 ± 0.887     | **0.258 ± 0.288**    | 0.248 ± 0.240 |
> | Pre Mult GN‑CDE              | 1.968 ± 0.548     | 12.262 ± 1.437    | 7.440 ± 4.458        | 6.829 ± 0.644        |
> | Original GN‑CDE (Qin et al.) | 0.366 ± 0.400     | 1.324 ± 0.630     | 0.417 ± 0.314 | 0.552 ± 0.470        |
> | **PENG‑CDE (ours)**          | **0.096 ± 0.051** | **0.481 ± 0.195** | 0.290 ± 0.265 | **0.247 ± 0.215**    |
> | **Relative Improvement**     | 73.84 %           | 63.70 %           | 30.44 %              | 55.20 %              |
>
>
> **Table 2: Comparison of GN-CDE variants and extended baselines on the gene regulation tasks. Mean MSEs with $95\%$ confidence intervals are reported, with the best mean highlighted in bold.**
> | Model                        | Community         | Grid                 | Power Law            | Small World          |
> | ---------------------------- | ----------------- | -------------------- | -------------------- | -------------------- |
> | **DCRNN**                        | 159.095 ± 149.970 | 28.784 ± 16.926      | 77.469 ± 28.178      | 27.300 ± 11.641      |
> | **STIDGCN**                      | 14.579 ± 2.815    | 0.633 ± 0.163        | 4.828 ± 0.694        | 0.611 ± 0.162        |
> | **ASTGCN**                       | 13.366 ± 3.302    | 0.695 ± 0.178        | 4.722 ± 0.644        | 0.579 ± 0.116        |
> | **STGNCDE**                  | 88.354 ± 15.126   | 8.753 ± 0.898        | 20.498 ± 2.633       | 6.760 ± 0.787        |
> | Constant                     | 36.307 ± 2.609    | 1.390 ± 0.168        | 7.099 ± 0.382        | 0.772 ± 0.126        |
> | Graph Neural ODE (GNODEs)    | 8.548 ± 3.212     | **0.167 ± 0.191**    | **0.372 ± 0.398**    | 0.294 ± 1.308 |
> | Adjacency GN‑CDE             | 8.909 ± 6.311     | 1.476 ± 2.504        | 0.498 ± 0.126 | 0.195 ± 0.046 |
> | Pre Mult GN‑CDE              | 153.084 ± 149.609 | 2.553 ± 0.251        | 6.978 ± 3.045        | 1.591 ± 0.146        |
> | Original GN‑CDE (Qin et al.) | 10.717 ± 7.079    | 0.457 ± 0.167        | 0.822 ± 0.299        | 0.323 ± 0.154 |
> | **PENG‑CDE (ours)**          | **4.566 ± 2.780** | 0.247 ± 0.090 | 0.526 ± 0.220 | **0.186 ± 0.484**    |
> | **Relative Improvement**     | 67.06 %           | 45.90 %              | 39.71 %              | 54.14 %
>
> **[W3] Minor grammatical error**
>
> Thank you for highlighting this. We have adjusted the paper accordingly.

---

> > ### Comment · Reviewer_Dvhd · 2025-08-04
> >
> > The feedback fulfils my concerns. I will increase my score.

---

> ### Author Response · Authors · 2025-08-06
>
> Dear Reviewer Dvhd,
>
> Thank you for your thoughtful consideration of our rebuttal and for adjusting your score. We are pleased that our response has addressed the points you raised. Should any concerns remain, we would be glad to provide further clarification.

---

### Official Review · Reviewer_GJRF · 2025-07-01

**Clarity:** 3
**Significance:** 3
**Originality:** 3
**Rating:** 5
**Confidence:** 3

**Summary:**

This paper introduces Permutation Equivariant Neural Graph Controlled Differential Equations (PENG-CDEs), a novel framework that extends Neural CDEs to dynamic graph settings while preserving permutation equivariance. The key innovation lies in projecting Graph Neural CDEs onto permutation equivariant function spaces, significantly reducing parameters without compromising expressiveness. The method demonstrates superior performance on both synthetic benchmarks and real-world tasks, particularly excelling in interpolation and extrapolation scenarios across various graph dynamics.

**Questions:**

How does the computational complexity of PENG-CDEs scale with graph size compared to traditional Graph Neural CDEs, especially in large-scale real-world applications?

Could you elaborate on the theoretical guarantees regarding the approximation power of the proposed equivariant function space projection?

What are the practical limitations when applying this method to highly sparse or rapidly evolving graph structures?

Have you considered extending this framework to handle discrete-time graph sequences or hybrid continuous-discrete dynamics?

**Ethical Concerns:**

["NO or VERY MINOR ethics concerns only"]

**Final Justification:**

PENG-CDE delivers consistent performance gains across benchmarks and provides clear theoretical grounding, although questions persist about its behavior on highly sparse or rapidly evolving graphs.

**Limitations:**

Yes

**Quality:**

3

**Strengths And Weaknesses:**

The paper presents a theoretically grounded permutation-equivariant neural CDE framework that demonstrates strong performance across synthetic and real-world benchmarks, achieving up to 73.84% improvement in predictive accuracy while maintaining computational efficiency through its parameter-reduced design. While the methodology shows impressive empirical results (outperforming baselines on TGB benchmarks by 5-67%), its reliance on dense adjacency matrices limits scalability to very large graphs, and the theoretical guarantees remain confined to linear cases. The work makes significant conceptual contributions by bridging geometric deep learning with neural differential equations, though broader impacts would benefit from validation on industrial-scale problems. The clear exposition of degeneracy modes and mitigation strategies enhances reproducibility, while the modular design enables straightforward extension to other equivariant architectures.

---

> ### Author Rebuttal · Authors · 2025-07-31
>
> We thank the reviewer for their time invested in reading the paper and for their constructive and helpful review.
>
> **[Q1] Computational complexity compared with original Graph Neural CDEs**
>
> We appreciate you highlighting this aspect of our model, since it was one of the key motivations for introducing PENG-CDEs.
>
> **Parameters:**  The number of parameters in the PENG-CDE is independent of the number of nodes  $n$  or edges  $e$, whereas in the original GN-CDE, the parameter count scales quadratically with  $n$. This makes the original formulation unsuitable for large-scale graph problems and negatively impacts runtime complexity.
>
> As shown in Table 1 below, even for a modest graph size of 2048 nodes, our model trains  **6.1× faster per epoch** on the experiments in section 4.1 compared to the original GN-CDE. We have now clarified these differences in Section 3.2.
>
> **Table1: Runtime per training epoch (s) for the GN-CDE and our PENG-CDE, for experiments in Section 4.1 for different graph sizes.**
> |Model\Num. of nodes | 128  | 256  | 512   | 1024 | 2048 |
> |--------|------|------|--------|------|-------|
> | (Pre Mult) GN-CDE    | 0.45 | 0.56 | 1.18  | 1.62 | 9.15  |
> | PENG-CDE (ours)   | 0.42 | 0.49 | 0.57| 0.90 | 1.50  |
>
>
>
> **[Q2] Theoretical guarantees for approximation power of equivariant projection**
>
> In terms of the statement of Theorem 3.1, we know that projecting an element in a vector space onto a subspace will give us the closest element in that subspace (see Appendix A for a more formal treatment of this). We use this notion to argue in Theorem 3.1 that (assuming the absence of non-linearities) PENG-CDEs are the best possible permutation equivariant approximation to the GN-CDE. Furthermore, [1] show that GN-CDEs are universal.
>
> **[Q3] Handling of rapidly evolving and sparse graph structures**
>
> Before training and inference, the provided graph snapshots are interpolated into a continuous driving path (which depends on the number of observations in the sequence). Once this preprocessing has been done, both the complexity of the forward and backward passes are decoupled from the number of input observations. Indeed, the number of vector field evaluations depends on the ODE solver rather than the sampling rate. This makes our model inherently robust to oversampled input data. For more details and experiments on this, we kindly refer the reviewer to Section 4.1 and Figure 2.
>
> When the graph structure itself evolves rapidly, any time series model will require a large number of observations to perform well. However, since the PENG-CDE is formulated as a controlled differential equation, it offers a distinct advantage over conventional time series models: it allows us to tap into the differential equations literature to understand how rapidly changing data will impact our model, by analysing the bounded variation norm of the input path.
>
> Regarding sparsity, we acknowledge that in its current implementation, just like the original GN-CDE, PENG-CDEs have quadratic memory complexity in the number of nodes due to the need to store dense adjacency matrices. This limitation, which we outline in the Limitations section, restricts scalability to problems with a large number of nodes. This constraint arises primarily from  JAX’s limited support for sparse matrix operations. With future versions of JAX moving its sparse support beyond the experimental stage, we will be able to address this  scalability bottleneck  is an important direction for future work. As shown in Table 1, once memory allocation constraints are met, PENG-CDE scales much more favorably in runtime compared to GN-CDE - demonstrating its potential for efficient execution, even at larger scales.
>
> We emphasise that for a fixed number of nodes, the memory footprint of PENG-CDEs is independent of the number of edges. This enables us to scale (within the same order of magnitude) to the largest dataset in the Temporal Graph Benchmark [2] in terms of the number of edges. For an overview of the edge counts in these datasets, we refer the reader to Figure 1 in [2].
>
> **[Q4] Extension to discrete-time graph sequences or hybrid continuous-discrete dynamics**
>
> By construction, CDEs - and by extension, PENG-CDEs - are well-suited to handle all these scenarios. While Sections 2 and 3 present the framework assuming continuous observations, discrete-time graph sequences can be naturally incorporated by interpolating the observations, for example using cubic splines. If the evaluations of the control path are ought to remain in a discrete domain, one can employ rectilinear interpolation, as discussed in [3].
> We note that all real-world experiments in Section 4.2 operate on discrete-time graph dynamics, which we handle precisely in this way, as described in Section 2.2. Hybrid continuous-discrete settings can also be accommodated straightforwardly: discrete-time regimes or features can be interpolated, while continuous ones are used directly.
> We agree that this point could be emphasised more clearly in the main paper and will add further details in Section 4.
>
> ---
> We hope to have addressed the reviewer’s questions and concerns, and we would be happy to engage in further discussion or respond to any follow-up questions.
>
> ---
>
> [1] Tiexin Qin, Benjamin Walker, Terry Lyons, Hong Yan, and Haoliang Li. Learning dynamic
> graph embeddings with neural controlled differential equations, 2023.
>
> [2] Shenyang Huang, Farimah Poursafaei, Jacob Danovitch, Matthias Fey, Weihua Hu, Emanuele Rossi, Jure Leskovec, Michael Bronstein, Guillaume Rabusseau, and Reihaneh Rabbany. Temporal graph benchmark for machine learning on temporal graphs. Advances in Neural Information Processing Systems, 2023
>
> [3] James Morrill, Patrick Kidger, Lingyi Yang, and Terry Lyons. On the choice of interpolation
> scheme for neural CDEs. Transactions of Machine Learning Research, 2022.

---

> > ### Comment · Reviewer_GJRF · 2025-08-07
> >
> > Thank you for the detailed responses. Please ensure the final version (1) includes the complexity table in the main paper, (2) explicitly discusses the limitations of rapidly-evolving or highly-sparse graphs, and (3) briefly highlights the discrete-time/hybrid extension as an open direction rather than a solved problem.

---

### Official Review · Reviewer_G4dS · 2025-07-02

**Clarity:** 4
**Significance:** 2
**Originality:** 3
**Rating:** 4
**Confidence:** 4

**Summary:**

This paper introduced a permutation equivariant graph neural controlled differential equation (PENG-CDEs) by leveraging 2nd-order linear mapping between the adjacency matrix (Equation 5). In synthetic and real-world datasets, it showed improvements over previous methods like GN-CDEs.

**Questions:**

1. For experiment 4.1, there's only one change in the graph structure in the validation and test phase. Is it enough to test the generalization?

2. In what scenarios is time-warp equivariance important? And why?

**Ethical Concerns:**

["NO or VERY MINOR ethics concerns only"]

**Final Justification:**

The questions I raised have been resolved. I still maintain my point about the paper; it's an improvement in graph neural CDE, however, the way it improves the original GN-CDE is a straightforward application of permutation-equivariant theory from standard graph neural networks. Given the potential impact this work might bring to the field of dynamic graph learning, I will weakly recommend accepting this paper, and I maintain my score of "4 Borderline Accept".

**Limitations:**

yes

**Quality:**

3

**Strengths And Weaknesses:**

**Strengths**:
1. Clarity. The method described in the paper is clear and sound.
2. Experiments. Experiments are relatively solid, with diverse settings and dataset and a strong choice of baselines. The improvement after adding the full linear equivariant functions over original GN-CDE is notable.

**Weakness**:
1. Significance: It is a pretty standard application of the well-known linear equivariant function w.r.t. $S_n$ to the dynamic graph setting. It only considered a simplification (Eq. 4) of the original graph neural CDE formulation (Eq. 3) and makes it equivariant w.r.t. node relabeling. This downgraded the significance and novelty of the proposed PENG-CDE.

---

> ### Author Rebuttal · Authors · 2025-07-31
>
> We thank the reviewer for their constructive and helpful review.
>
> **[W1]: Limited Significance**
>
> Although the permutation-equivariant linear map we employ is well-known, we do not believe our contribution is limited to a routine application. Theorem 3.1 proves that projecting the GN-CDE vector field onto the equivariant subspace is the optimal choice, establishing new theory for dynamic-graph CDEs. Empirically, this projection delivers large practical gains: on our synthetic benchmarks, PENG-CDE uses only 2% more parameters than GN-CDE (see Table 1 below) yet reduces average MSE by between 30.44% and 73.84%. Compared to the Pre-Mult GN-CDE variant, PENG-CDE reduces the parameter count by over 99%, while still achieving 8.56× to 27.65× lower average MSE. Finally, our method belongs to a family of models that are unique in their approach to tackling graph dynamics. Most other approaches are permutation equivariant due to handling the graph structure using an existing graph neural network structure. Therefore, introducing the first permutation equivariant graph neural CDE represents an important step for this class of models. We believe this represents a meaningful advancement in the design of continuous-time models for dynamic graphs.
>
> **Table 1: Comparison of number of parameters for GN-CDE, its Pre Mult version and PENG-CDE on our synthetic experiments**
> | Model           | No. of parameters |
> | --------------- | ----------------- |
> | GN-CDE          | 1,741             |
> | PENG-CDE (ours) | 1,771             |
> | Pre Mult GN-CDE | 641 k             |
>
>
> **[Q1]:  Change in Graph Structure during Validation**
>
> In all synthetic experiments in Section 4.1, the underlying graph topology changes at 12 uniformly random snapshots out of the total 120 time steps. This implies that the expected number of topology changes in the validation window (20 snapshots for interpolation validation, 20 for extrapolation) is 40 / 120 × 12 = 4. As shown in Figure 1, where vertical gray dashed lines indicate the timing of topology changes, 3 changes occur within the extrapolation regime of the validation set in this setting. We consider this number sufficient to assess the model’s ability to generalise to evolving graph structures. For testing, we generate entirely new time series realisations, each containing 12 randomly sampled topology changes. This ensures that the test set evaluates generalisation to unseen sequences and topological dynamics. We have modified Section 4 to make this clearer.
>
> **[Q2]:  Importance of time-warp equivariance**
> Time-warp equivariance is important in time series modeling when the task depends on the sequential structure of the signal rather than the absolute timing of events. For example, in classifying whether a trajectory draws the digit “6”, it is irrelevant how fast or slowly it is traced, as long as the shape is preserved. Similarly, if you consider classifying the nodes of a social media network as to whether they are friends with a specific node or friends-of-a-friend with that node. The important characteristic is the order in which the edges appeared (you can't connect with a friend-of-a-friend without first connecting with the friend) and not the time in-between those connections appearing. We will emphasise the motivation for this type of equivariance more clearly in the camera-ready version, as well as detailing how you can avoid it by simply including time as a channel in the driving path when you believe that your problem is not time-warp equivariant.

---

> > ### Comment · Reviewer_G4dS · 2025-08-06
> >
> > Dear authors,
> >
> > Thank you for the explanations. I understand that PENG-CDE is the first permutation-equivariant graph neural CDE. Also, I noted the improvement with the architecture in both synthetic and real-world datasets. However, the improvement wasn't consistent in some cases, like in the Power Law dataset regarding the gene regulation task and in the Twitter-Tennis dataset where the improvement is pretty minor. According to my experience, when you naively substitute a full-rank transformation for an equivariant map (in this paper's case, in Eq. 5), it will sometimes bring large improvement but sometimes not. Besides, I still think the use of equivariant fusion operator (or the projection) here is pretty straightforward, as the only permutation equivariant component in Eq. 4 is the adjacency matrix (and its derivative), the straightforward and the only way to make it permutation equivariant is to do a permutation equivariant map on them.
> >
> > Nonetheless, I recognize the potential impact of the new PENG-CDE architecture might bring in the field of dynamic graph learning. So I plan to maintain my score "4 Borderline Accept" unless any significant misunderstanding is raised.

---

> > > ### Author Response · Authors · 2025-08-06
> > >
> > > Dear Reviewer G4dS,
> > >
> > > Thank you for recognising the impact of our work and recommending our paper for acceptance.

---

### Note · Authors · 2025-08-14

We sincerely thank the reviewers and AC for their time and consideration. We are encouraged that the paper was generally well received, with reviewers describing our extension of Neural CDEs to dynamic graph settings while preserving permutation equivariance as *"novel"* (GJRF, Dvhd) and recognising that our work *"makes significant conceptual contributions by bridging geometric deep learning with neural differential equations"* (GJRF).

The *"robust theoretical foundation"* (Dvhd) was noted as *"rigorous and detailed"* (Dvhd), and our empirical results were described as *"impressive"* (GJRF), achieving a new *state-of-the-art on TGB-genre* (FW39). We are also pleased that the paper was considered *"clearly written"* (Dvhd) and *"sound"* (G4dS) presentation.


In the revised final version, we have incorporated the reviewers' comments and made the following key improvements:

- Clarified in Section 4.1 how the validation set is constructed, highlighting that it contains on average four graph topology changes.
- Strengthened the motivation for time-warp equivariance, and explained how it can be disabled by including time as a channel in the driving path when it is not desired (Section 2).
- Added runtime complexity comparisons between GN-CDE and PENG-CDE (see our response to reviewer GJRF, Section 4).
- Emphasised how PENG-CDEs natively handle discrete-time and hybrid continuous–discrete dynamics (Section 4).
- Added a discussion of the limitations of time-series models for rapidly-evolving or highly-sparse graphs.
- Included additional baselines from Section 4.2 into the synthetic experiments in Section 4.1 (response to Dvhd) and added two more dynamical systems (response to FW39).
- Integrated critical difference diagrams to study the statistical significance if our results from our response to FW39 into Section 4 and the Appendix.
- Added a mechanistic discussion of the ablation study in Table 9.
- Moved information about search spaces for hyperparameters from the Appendix into the main text for improved readability.


We thank the reviewers again for their constructive feedback, which has substantially improved the quality of our work. We hope that with these modifications and our detailed individual responses, we have addressed all concerns.

---

### Decision · Program_Chairs · 2025-09-17

**Decision:**

Accept (poster)

**Comment:**

The paper introduces a permutation equivariant model for controlled differential equations to enable representation learning on dynamic graphs. The key idea of the model is to leverage permutation equivariant function spaces to reduce the number of parameters while preserving model capacity, leading to more efficient training and better generalization. The work has several clear merits highlighted by the reviewers, including the solid formulation of the model and the detailed experimental results. During the discussion period, the authors provided clarifications and addressed several of the reviewers' concerns. Overall, the paper makes a good contribution to the field. I would encourage the authors to take into account the reviewers' comments to further improve clarity.